# STRIDE: Post-Training LLMs to Reason and Refine Bio-Sequences via Edit Trajectories

**Daiheng Zhang** [1]  **Shiyang Zhang** [2]  **Sizhuang He** [2]  **Yangtian Zhang** [2]  **Syed Asad Rizvi** [2]  **David van Dijk** [2]

## Abstract

Discrete biological sequence optimization often requires goal-directed, parser-valid edits to an existing protein or molecule. Diffusion models support iterative refinement but do not expose a controllable discrete-edit interface, while autoregressive LLMs can be myopic when planning constrained edits over multiple steps. We introduce *STRIDE* (Sequence Trajectory Refinement via Iterative Discrete Editing), a post-training framework that trains an LLM to emit executable INSERT/DELETE/REPLACE trajectories for variable-length refinement. *STRIDE* first learns Levenshtein-aligned shortest-edit demonstrations, then uses supervised fine-tuning and group-based policy optimization to align trajectories with task rewards while preserving coherent editing. On an oracle-based full-action protein stress test, *STRIDE* raises success over Vanilla SFT from 42% to 89% and novelty among unique improvements from 47% to 97%. On instruction-conditioned molecular editing, the GSPO-aligned variant improves strict success, controllability, and SMILES validity over the SFT-only *STRIDE* model (code: https://github.com/daiheng-zhang/STRIDE).

## 1. Introduction

Designing and optimizing biological sequences—including proteins and small molecules—is central to computational biology, with applications spanning therapeutics, enzyme engineering, and materials discovery (Yang et al., 2019). In many realistic settings, the goal is not de novo generation but goal-directed refinement (Arnold & Volkov, 1999; Zhou et al., 2019): starting from a viable precursor and applying a small number of edits that improve a target property while improving adherence to syntactic and structural constraints under parser-based validation (e.g., amino-acid constraints or SMILES grammar (Weininger, 1988)). This yields a challenging search problem in an enormous discrete space with hard constraints (Romero & Arnold, 2009) and often variable-length transformations.

Recent generative paradigms offer complementary strengths for this refinement setting. Diffusion models provide a powerful template for iterative improvement through progressive denoising (Ho et al., 2020; Song et al., 2021), and have been extended to categorical token spaces via discrete diffusion processes (Austin et al., 2021; Hoogeboom et al., 2022). In biology, discrete diffusion models have shown strong generative ability in protein sequence space (Alamdari et al., 2023; Wang et al., 2024). However, when the desired control interface is an explicit edit policy—especially under insertions and deletions—these approaches typically rely on specialized transition parameterizations and sampling procedures, making it difficult to (i) enforce domain-specific validity at every intermediate step and (ii) keep edits interpretable and directly controllable (Gu et al., 2019).

Autoregressive large language models (LLMs) (Brown et al., 2020), by contrast, natively operate over discrete tokens and can be adapted to generate structured sequence representations. Yet, when repurposed for optimization, standard decoding can be myopic (Yao et al., 2023a): locally plausible edits do not necessarily realize the long-horizon plans needed to navigate rugged fitness landscapes under tight edit budgets. This tension motivates a method that retains the iterative, multi-step character of refinement while leveraging the priors of token-level generation.

We introduce *STRIDE* (**S**equence **T**rajectory **R**efinement via **I**terative **D**iscrete **E**diting), a post-training framework that reformulates discrete sequence optimization as trajectory planning in edit space. Rather than learning a separate stochastic transition process, *STRIDE* trains an LLM to emit an explicit trajectory of atomic edits (INSERT/DELETE/REPLACE) that progressively transforms a source sequence into an optimized candidate. These trajectories provide a transparent control interface

[1]Department of Electrical and Computer Engineering, Rutgers University [2]Department of Computer Science, Yale University. Correspondence to: Daiheng Zhang <dz367@rutgers.edu>, David van Dijk <david.vandijk@yale.edu>.

*Proceedings of the 43$^{rd}$ International Conference on Machine Learning*, Seoul, South Korea. PMLR 306, 2026. Copyright 2026 by the author(s).

for variable-length editing while keeping each step interpretable.

*STRIDE* follows a two-stage training recipe. Stage I performs supervised fine-tuning on shortest edit-path demonstrations derived from Levenshtein alignment (Levenshtein, 1966), initializing a validity-oriented editor. Stage II aligns edit trajectories with task rewards using group-based policy optimization (e.g., GRPO and variants), with KL regularization to preserve coherent editing behavior. We evaluate *STRIDE* on protein optimization benchmarks spanning GFP fluorescence (TAPE) (Rao et al., 2019) and an AAV transfer evaluation (Dallago et al., 2021; Bryant et al., 2021), together with instruction-conditioned molecular editing benchmarks (Fernandez et al., 2025; Ye et al., 2025), demonstrating improved optimization success and substantially stronger diversity/novelty, with gains most pronounced in variable-length, index-consistent editing regimes. Our contributions are:

- **Executable, variable-length edit trajectories.** We cast discrete bio-sequence refinement as planning over executable atomic edits (INSERT/DELETE/REPLACE) for index-consistent, variable-length editing.

- **Trajectory supervision via Levenshtein backtracing.** A deterministic DP pipeline converts aligned pairs into index-grounded edit scripts for SFT, inducing a minimal-edit, validity-biased prior.

- **Reward-aligned post-training for edit scripts.** Group-based policy optimization with KL regularization aligns trajectories with task rewards under parse-and-execute consistency.

- **Empirical gains under controllability constraints.** On GFP/AAV, *STRIDE* improves success and diversity/novelty; on instruction-conditioned molecule editing, it improves validity and reduces off-target property shifts, especially in variable-length regimes.

**Conflict of Interest Disclosure.** The authors declare no financial conflicts of interest. None of the models, datasets, or services evaluated in this work are products of companies that employ any of the authors.

## 2. Related Work

### 2.1. Discrete Diffusion for Sequence Modeling

Discrete diffusion extends diffusion modeling to categorical token spaces, enabling direct corruption and denoising of sequences. In natural language processing, representative approaches include fully discrete token diffusion (e.g., D3PM (Austin et al., 2021)), continuous-latent text diffusion

(e.g., Diffusion-LM (Li et al., 2022)), and autoregressive–diffusion hybrids such as Block Diffusion (Arriola et al., 2025). In biology, discrete diffusion models have been applied to protein sequence generation, including EvoDiff (Alamdari et al., 2023) and DPLM (Wang et al., 2024), building on pre-trained protein language models such as ESM (Rives et al., 2021). However, these approaches predominantly target fixed-length generation and do not naturally expose an explicit, step-by-step *executable* edit policy under variable-length operations. In contrast, edit-based sequence modeling explores insertion/deletion dynamics via explicit edit processes, such as CTMC-based insertion/deletion/replacement in Edit Flows (Havasi et al., 2025) and mask-insertion paradigms such as FlexMDMs (Kim et al., 2025). Our work connects these lines by training an autoregressive LLM to emit verifiable trajectories of atomic edits for controllable optimization.

### 2.2. Reasoning-Oriented Post-Training

Modern post-training typically couples supervised fine-tuning (SFT) on high-quality demonstrations with preference optimization to align models with human judgments. For complex tasks, reasoning-oriented supervision (Wei et al., 2022; Chung et al., 2024) can improve multi-step deduction by training models to produce intermediate reasoning traces. InstructGPT (Ouyang et al., 2022) established that SFT followed by reinforcement learning from human feedback (RLHF) can substantially improve human-rated quality. For reasoning-centric tasks, Group Relative Policy Optimization (GRPO) (Shao et al., 2024; Guo et al., 2025) has emerged as an efficient alternative to PPO-style RLHF without a learned critic. Recent variants further improve stability and efficiency, including Group Sequence Policy Optimization (GSPO) (Zheng et al., 2025) and Clipped Importance Sampling Policy Optimization (CISPO) (Chen et al., 2025), which highlight different stability–plasticity trade-offs when optimizing long sequences.

### 2.3. Reasoning Traces for Bio-Sequence and Molecular Design

Recent work in scientific generation increasingly moves beyond outcome-only supervision by eliciting or supervising explicit reasoning traces, improving interpretability and enabling structured alignment. In peptide and molecular design, PepThink-R1 (Wang et al., 2025) combines chain-of-thought (CoT) supervised fine-tuning with reinforcement learning to produce interpretable rationales over monomer-level modifications for cyclic peptide optimization, while Mol-R1 (Li et al., 2025) targets text-based molecule discovery and improves explicit long-CoT reasoning via distillation and iterative SFT/RL. In genomics, BioReason (Fallahpour et al., 2025) couples a DNA foundation model with an LLM and trains multi-step biological deduction traces

using a curriculum of supervised and reinforcement learning. Complementary to purely verbal traces, tool-augmented chemistry agents such as ChemCrow (M. Bran et al., 2024) ground intermediate reasoning by interleaving thoughts with executable tool calls, and general paradigms like ReAct (Yao et al., 2023b) similarly emphasize action-anchored or executable reasoning traces. Collectively, these efforts motivate explicit process traces for scientific controllability.

## 3. Method

This section presents *STRIDE*, which formulates discrete biological sequence optimization as trajectory planning in edit space. *STRIDE* generates an explicit sequence of atomic edits INSERT/DELETE/REPLACE that progressively refines an initial sequence, using a *Shortest Edit Path* (SEP) derived from Levenshtein alignment with dynamic-programming backtracking as supervision. Training follows a two-stage recipe: (i) supervised fine-tuning to model both edit trajectories and final sequences, encouraging validity and a minimal-edit bias; and (ii) GRPO-style post-training to align trajectories with task rewards, with KL regularization to preserve coherent edit behavior. This design yields interpretable and controllable refinement trajectories while improving downstream optimization performance.

### 3.1. Constructing Optimal Edit Trajectories via Dynamic Programming

Given a source token sequence $x_{\text{src}} = (x_1^{\text{src}}, \ldots, x_m^{\text{src}})$ (e.g., a wild-type protein or an initial molecule) and a target token sequence $x_{\text{tgt}} = (x_1^{\text{tgt}}, \ldots, x_n^{\text{tgt}})$ (e.g., a higher-fitness variant), we seek a minimum-cost edit script $T^\star = (a_1, \ldots, a_N)$ that transforms $x_{\text{src}}$ into $x_{\text{tgt}}$ under unit-cost INSERT/DELETE/REPLACE operations. We define the atomic action set $\mathcal{A} = \{\text{INSERT}, \text{DELETE}, \text{REPLACE}\}$.

**Tokenization and structural prior.** We operate on token sequences: amino-acid tokens for proteins and a Regex SMILES tokenizer for molecules. We use the shortest edit path implied by Levenshtein alignment as a conservative structural prior, biasing the model toward local, minimal modifications rather than global rewrites.

**Executable trajectory and indexing.** We interpret an edit trajectory as an executable program applied to the evolving sequence. At step $t$, an action $a_t = (op_t, p_t, v_t)$ is applied to the current sequence $x_{t-1}$ of length $L_{t-1}$, where positions are *0-based* and always interpreted with respect to $x_{t-1}$ (not the initial $x_{\text{src}}$). Concretely: INSERT$(p, v)$ inserts token $v$ before position $p$ for $0 \leq p \leq L_{t-1}$ (with $p = L_{t-1}$ appending at the end); DELETE$(p)$ removes $x_{t-1}[p]$ for $0 \leq p < L_{t-1}$; and REPLACE$(p, v)$ sets $x_{t-1}[p] \leftarrow v$ for $0 \leq p < L_{t-1}$. Because indices are re-evaluated after

*Table 1.* Sample inference on a molecular optimization task.

**User prompt**
Can you make molecule CC(=O)Nc1cc(NC(=O)N[C@@H](C CO)c2cccs2)ccc1C more like a drug? The output molecule should be similar to the input molecule.
Please return:
- edit_traj (using only INSERT/DELETE/REPLACE)
- output_smiles

**Edit trajectory**
<edit_traj>
DELETE(21)
REPLACE(31, 2)
INSERT(7, =)
DELETE(11)
INSERT(8, C)
...
</edit_traj>

**Answer**
CC(=O)N=Cc1ccc(C=NC(=O)N[C@@H](CCO)c2cccs2)cc1

each operation, the script is unambiguous when executed sequentially.

**Levenshtein DP and backtracing.** We compute the Levenshtein dynamic-programming table $D \in \mathbb{Z}_{\geq 0}^{(m+1) \times (n+1)}$ with base cases $D[i, 0] = i$ and $D[0, j] = j$, and recurrence

$$D[i, j] = \min \Big\{ D[i-1, j] + 1, \ D[i, j-1] + 1,$$
$$D[i-1, j-1] + \mathbb{I}[x_i^{\text{src}} \neq x_j^{\text{tgt}}] \Big\}.$$

Starting from $(i, j) = (m, n)$, we backtrace to obtain a minimum-cost alignment path. To convert this alignment into an *executable* script with dynamic positions, we replay the alignment steps forward on a mutable copy of $x_{\text{src}}$ while maintaining a cursor in the current sequence; insert/delete operations update both the sequence and cursor so that emitted positions always refer to the current state. When multiple predecessors tie during backtracing, we break ties deterministically (fixed priority order) for reproducibility.

**Serialization for training.** We serialize the resulting edit script $T^\star$ as a structured edit trace followed by the target output: $(x_{\text{src}}, I) \rightarrow \langle\text{edit\_traj}\rangle T^\star \langle/\text{edit\_traj}\rangle \rightarrow x_{\text{tgt}}$. Note that while endpoints are valid sequences, intermediate strings along the shortest edit path are not guaranteed to satisfy domain-specific validity (e.g., SMILES well-formedness); we use the script as process supervision and evaluate validity on the final output.

### 3.2. Stage I: Valid and Minimal Editing via SFT

**Training format.** For each training pair $(x_{\text{src}}, x_{\text{tgt}})$ and instruction $I$, we compute a shortest edit script $T^\star$ using the DP backtracing procedure in Section 3.1. We then form a

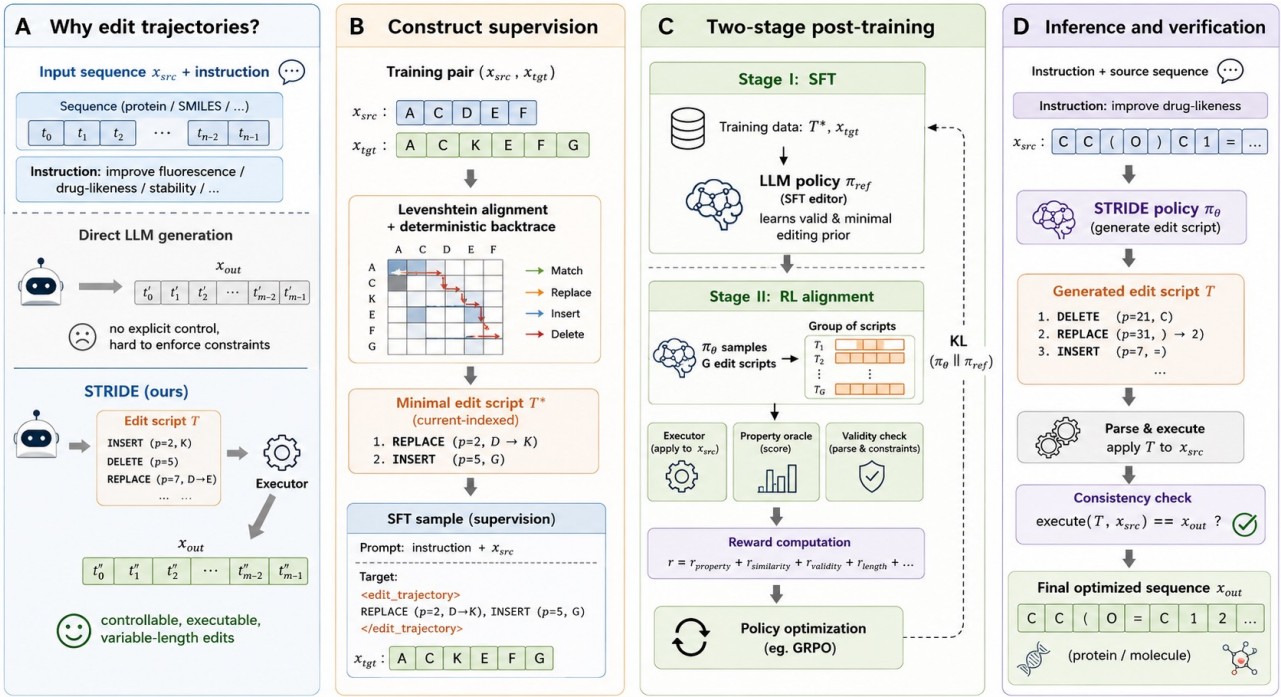

*Figure 1.* Overview of the *STRIDE* workflow. (A) *STRIDE* optimizes biological sequences through executable edit scripts rather than unconstrained direct generation. (B) Supervised edit trajectories are constructed from Levenshtein alignment with deterministic backtracking. (C) The editor is first trained with SFT and then aligned with GRPO-style rewards, validity checks, and KL regularization. (D) At inference, the policy generates an edit script that is parsed, executed, and verified before producing the final optimized sequence.

prompt $q = (x_{\text{src}}, I)$ and a completion

$$y = \big[ \langle\texttt{edit\_traj}\rangle \, T^\star \, \langle\texttt{/edit\_traj}\rangle \, ; \, x_{\text{tgt}} \big],$$

i.e., the model is trained to first emit an explicit edit trajectory and then the final target sequence.

**Objective.** We perform standard teacher-forced supervised fine-tuning by maximizing the likelihood of the completion tokens. We minimize the negative log-likelihood

$$\mathcal{L}_{\text{SFT}}(\theta) = -\mathbb{E}_{(q,y)\sim\mathcal{D}} \sum_{t=1}^{|y|} \log \pi_\theta(y_t \mid q, y_{<t}),$$

where $\pi_\theta$ is a causal language model and the loss is computed over $y$ (trajectory + final output). We denote the resulting SFT policy as $\pi_{\text{ref}}$ and use it as the KL reference in Stage II.

**What SFT internalizes.** Supervising $T^\star$ (process) together with $x_{\text{tgt}}$ (outcome) provides two inductive biases: (i) *validity-biased prior*—training targets are valid endpoints (proteins/SMILES), so the model learns an implicit prior for producing valid sequences without constrained decoding; validity is evaluated post hoc (e.g., RDKit for SMILES); (ii) *minimal-edit bias*—because $T^\star$ is the unit-cost shortest script, the model is encouraged to produce concise, non-redundant trajectories rather than convoluted chains.

### 3.3. Stage II: Functional Alignment via Group Relative Policy Optimization (GRPO)

Stage I trains a conservative, validity-oriented editor by imitating shortest edit paths. Stage II further aligns the editor with task objectives by optimizing task rewards while regularizing to the Stage I reference policy $\pi_{\text{ref}}$.

**Rollouts and rewards.** Given a prompt $q = (x_{\text{src}}, I)$, the policy $\pi_\theta$ generates a completion $o = [\langle\texttt{edit\_traj}\rangle T \langle\texttt{/edit\_traj}\rangle; x_{\text{out}}]$ containing an edit trajectory $T$ and a final sequence $x_{\text{out}}$. We compute a scalar reward $r = R(x_{\text{out}}; x_{\text{src}}, I)$ using task oracles. To ensure we truly align trajectories, if $T$ is not parsable/executable or executing $T$ on $x_{\text{src}}$ does not reproduce $x_{\text{out}}$, we set $r = 0$.

**Group Sampling and Relative Advantage.** GRPO optimizes $\pi_\theta$ without a learned critic. For each prompt $q$, we sample a group of $G$ outputs $\{o_i\}_{i=1}^{G} \sim \pi_{\theta_{\text{old}}}(\cdot \mid q)$ and compute rewards $\{r_i\}_{i=1}^{G}$. We then form a group-

normalized advantage

$$A_i = \frac{r_i - \mu_r}{\sigma_r + \epsilon}, \qquad \mu_r = \frac{1}{G}\sum_{j=1}^{G} r_j,$$

$$\sigma_r = \sqrt{\frac{1}{G}\sum_{j=1}^{G}(r_j - \mu_r)^2}.$$

Since our reward is outcome-level (defined on the final sequence), we assign the same $A_i$ to all tokens in $o_i$.

**GRPO objective.** We optimize $\pi_\theta$ using a PPO-style clipped surrogate with KL regularization to $\pi_{\text{ref}}$:

$$J_{\text{GRPO}}(\theta) = \mathbb{E}\left[\frac{1}{G}\sum_{i=1}^{G}\frac{1}{|o_i|}\sum_{t=1}^{|o_i|}\right.$$

$$\min\left(\rho_{i,t}A_i,\ \text{clip}(\rho_{i,t}, 1-\varepsilon, 1+\varepsilon)A_i\right)$$

$$\left. -\ \beta\, D_{\text{KL}}\big(\pi_\theta(\cdot \mid q)\,\|\,\pi_{\text{ref}}(\cdot \mid q)\big)\right].$$

where $\rho_{i,t} = \frac{\pi_\theta(o_{i,t}|q,o_{i,<t})}{\pi_{\theta_{\text{old}}}(o_{i,t}|q,o_{i,<t})}$. In practice we minimize $\mathcal{L}_{\text{GRPO}}(\theta) = -J_{\text{GRPO}}(\theta)$. Intuitively, the advantage term reinforces trajectories that yield higher task scores, while the KL term anchors the policy to the valid editing logic learned in Stage I, mitigating reward hacking and degeneration.

**Reward design.** **Protein (fluorescence) optimization.** With a fixed fluorescence oracle $f_{\text{fl}}(\cdot)$ and edit count $d(x_{\text{src}}, x_{\text{out}})$ (length of the executed script), we define an edit-budget and an improvement indicator:

$$\mathbb{I}_{\text{edit}} = \mathbb{I}[1 \leq d(x_{\text{src}}, x_{\text{out}}) \leq 3],$$
$$\mathbb{I}_{\text{fl}} = \mathbb{I}[f_{\text{fl}}(x_{\text{out}}) > f_{\text{fl}}(x_{\text{src}})].$$

The reward is $R_{\text{protein}} = \mathbb{I}_{\text{edit}} + \mathbb{I}_{\text{fl}}$.

**Molecular optimization.** We use RDKit to check validity. Let $\mathbb{I}_{\text{valid}}$ indicate successful parsing/sanitization. For target-property satisfaction, we assign a discrete score $R_{\text{prop}} \in \{0, 0.5, 1\}$ based on whether the generated molecule meets loose/strict thresholds under instruction $I$. To preserve structure, we compute Tanimoto similarity $S$ between input and output using Morgan fingerprints and set $R_{\text{sim}} \in \{0, 0.5, 1\}$ according to similarity thresholds. To control off-target drift, we apply a penalty $R_{\text{stable}} \leq 0$ if any non-target property change exceeds preset margins. The final reward is

$$R_{\text{mol}} = \big(\mathbb{I}_{\text{valid}} \cdot R_{\text{prop}} \cdot R_{\text{sim}}\big) + R_{\text{stable}}.$$

**GSPO and CISPO variants.** We also evaluate two recent variants of GRPO under the same trajectory reward, parse-execute consistency check, and KL anchor to $\pi_{\text{ref}}$.

GSPO (Zheng et al., 2025) shifts importance sampling from the token level to the sequence level, reducing the variance of token-wise updates over long edit trajectories. CISPO (Chen et al., 2025) clips the importance-sampling weights directly rather than the policy ratio. Both preserve the group-normalized advantage and reference-policy KL term; only the clipping/weighting mechanism differs from the GRPO surrogate above.

## 4. Experiments and Results

### 4.1. Datasets

We evaluate *STRIDE* across two biological sequence optimization settings: (i) protein sequence optimization, with GFP fluorescence as the main controlled benchmark and AAV capsid viability/packaging as a cross-protein transfer evaluation, and (ii) instruction-conditioned molecular optimization. In both domains, we train with the official train/validation splits provided by the underlying datasets unless stated otherwise.

#### 4.1.1. PROTEIN SEQUENCE OPTIMIZATION.

We use GFP fluorescence as the primary protein benchmark and AAV capsid viability/packaging as a cross-protein transfer setting. For GFP, we use the Fluorescence Landscape Prediction task from TAPE (Rao et al., 2019), which is derived from large-scale mutagenesis of *Aequorea victoria* GFP (avGFP) with experimentally measured log-fluorescence labels (Sarkisyan et al., 2016). Following TAPE, the split is defined by amino-acid Hamming distance to the wild-type: training variants are within distance $\leq 3$ and test variants are at distance $\geq 4$.

**SFT subset construction.** We cast optimization as goal-directed editing from a fixed wild-type anchor $x_{\text{src}}$ with label $y_{\text{src}}$. From the TAPE train/validation split, we keep only beneficial variants $x_i$ with $\Delta y_i = y_i - y_{\text{src}} > 0$ and construct paired supervision $(x_{\text{src}}, x_i)$. Each pair is converted into an edit-trajectory supervision signal using the shortest-edit backtracing procedure described in Section 3.1. After filtering, we obtain 3,280 training and 785 validation examples for SFT.

**Synthetic indel augmentation.** The TAPE fluorescence dataset only consists of substitution-only variants and provides limited coverage for insertions/deletions. To expose the editor to variable-length operations, we sample 1–3 random atomic edits $\mathcal{A}$ (INSERT/DELETE/REPLACE) applied to $x_{\text{src}}$ and assign pseudo labels using a fixed fluorescence predictor (SaProtHub/Model-Fluorescence-650M) (Su et al., 2025). We keep pseudo-improved samples with $\Delta y_i > 0$, yielding 7,153 training and 1,789 validation examples.

**AAV transfer evaluation.** For cross-protein transfer (Table 4b), we use the FLIP AAV capsid landscape (Dallago et al., 2021) from Bryant et al.'s AAV2 VP1 28-aa mutational screen (positions 561–588) with measured viability/packaging labels (Bryant et al., 2021); we use the sampled split (66,066 train / 16,517 test), the AAV2 WT segment anchor, and the same full-action protocol as GFP.

### 4.1.2. MOLECULAR OPTIMIZATION.

For molecules, we use the MEGA-MolEdit-522K dataset (Fernandez et al., 2025), which provides instruction-conditioned SMILES editing examples annotated with full atomic operation types (replace/insert/delete) across 14 optimization conditions (single- and dual-objective). Although MEGA includes edit annotations, our model operates on index-grounded token-level trajectories; we therefore convert each pair $(x_{\mathrm{src}}, x_{\mathrm{tgt}})$ into an executable index-level edit trajectory via Levenshtein alignment and backtracing. We use the positive splits for SFT and reporting (train/validation).

**GRPO training subset and evaluation set.** For GRPO, we sample 10,000 training examples from the MEGA-MolEdit train split with balanced coverage over the 14 conditions. We evaluate on 500 molecules from the DrugAssist (MolOpt-Instructions) test set (Ye et al., 2025), and perform cross-evaluation by optimizing each test molecule under all 14 conditions. We canonicalize SMILES and remove any overlaps between the evaluation set and MEGA-MolEdit training molecules to avoid data leakage.

## 4.2. Baselines

To evaluate trajectory-based atomic editing under a controlled budget, we benchmark *STRIDE* against baselines that isolate: (i) non-informed perturbations, (ii) direct sequence generation without explicit edit trajectories, and (iii) domain-specific generative models. Unless otherwise stated, all LLM-based baselines use the same Qwen3-14B backbone for controlled comparisons.

### 4.2.1. GENERAL BASELINES

**Random Perturbation.** A stochastic lower-bound that applies 1–3 random edit operations sampled uniformly from the action space $\mathcal{A}$ to $x_{\mathrm{src}}$. For molecules, random edits may yield invalid SMILES; we keep such outputs as invalid (counted in validity) and do not repair them.

**Zero-Shot LLM.** The base Qwen3-14B model is prompted to directly generate the optimized sequence conditioned on $(x_{\mathrm{src}}, I)$, without any parameter updates. This baseline measures the base model's prior capability for domain-valid generation and instruction following.

**Vanilla SFT (Direct, No-Traj).** An ablation where the model is supervised to directly predict the target sequence from the source (i.e., $(x_{\mathrm{src}}, I) \to x_{\mathrm{tgt}}$), without emitting an explicit executable edit trajectory. This isolates the benefit of explicit trajectory supervision from standard outcome-only imitation.

**Vanilla GSPO (Direct RL, No-Traj).** Starting from the Vanilla SFT checkpoint, we apply sequence-level policy optimization to directly maximize the same task rewards, still without explicit edit trajectories. This baseline isolates the gains from RL post-training alone, decoupled from trajectory generation.

### 4.2.2. TASK-SPECIFIC GENERATIVE BASELINES

For protein optimization, we compare against discrete generative sequence models that are not based on explicit edit scripts. **Discrete Diffusion (EvoDiff).** We fine-tune the EvoDiff-38M checkpoint on the same protein training set. Because EvoDiff is substitution-centric, we evaluate it in a *replace-only* setting for a fair comparison. **Edit Flow.** Edit Flow inherently supports variable-length operations. We therefore evaluate it under two regimes: (i) *replace-only* for a direct comparison with EvoDiff, and (ii) the full action space $\mathcal{A}$ to assess variable-length editing. Implementation details are provided in Appendix A.

## 4.3. Evaluation Metrics

**Protein Evaluation Metrics.** We evaluate protein optimization using a task-specific scalar score $s_{\mathrm{prot}}(\cdot)$: for GFP, $s_{\mathrm{prot}} = f_{\mathrm{fl}}$, the fixed fluorescence oracle; for AAV, $s_{\mathrm{prot}}$ is the viability/packaging landscape score. For each method, we sample $N = 100$ candidates $\{x_{\mathrm{out}}^{(k)}\}_{k=1}^{N}$ from the same source sequence $x_{\mathrm{src}}$ (and the same action-space setting, e.g., replace-only vs. full $\mathcal{A}$). We define the improved set $\mathcal{S}^+ = \{x_{\mathrm{out}}^{(k)} \mid s_{\mathrm{prot}}(x_{\mathrm{out}}^{(k)}) > s_{\mathrm{prot}}(x_{\mathrm{src}})\}$. We report: (i) **Success**: $|\mathcal{S}^+|/N$ (shown as $a/N$ in tables); (ii) **Uniqueness**: the fraction of distinct sequences among improved samples, i.e., $|\mathrm{Unique}(\mathcal{S}^+)|/|\mathcal{S}^+|$ (shown as $b/a$); (iii) **Novelty**: the fraction of unique improved sequences not appearing in the positive SFT training set $\mathcal{D}^+$, i.e., $|\{x \in \mathrm{Unique}(\mathcal{S}^+) : x \notin \mathcal{D}^+\}|/|\mathrm{Unique}(\mathcal{S}^+)|$ (shown as $c/b$). All set membership tests are exact string matches on amino-acid sequences.

**Molecule Evaluation Metrics.** We evaluate instruction-conditioned molecular editing across 14 conditions on 500 source molecules (i.e., 7,000 instances in total). Each instance specifies a target subset of proxy properties $\mathcal{P}_I \subseteq \mathcal{P}$ and desired directions, where $\mathcal{P} = \{\mathrm{LogP}, \mathrm{QED}, \mathrm{TPSA}, \mathrm{HBA}, \mathrm{HBD}\}$.

(i) **Validity.** Given an output SMILES $x_{\mathrm{out}}$, we first com-

*Table 2.* Per-task validity, success, and non-target property shift on DrugAssist for *STRIDE*-SFT vs. *STRIDE*-GSPO; rows are the 14 optimization tasks. Shift-related metrics are lower-is-better; we bold the better value (higher for Valid/Success, lower for Shift) between SFT and GSPO per task. "–" marks target properties (excluded from non-target shift). For per-property breakdowns see Appendix D.

| Task | Valid ↑ | | Success (Strict) ↑ | | Success (Loose) ↑ | | Shift Rate ↓ | | Shift Avg ↓ | |
|---|---|---|---|---|---|---|---|---|---|---|
| | SFT | GSPO | SFT | GSPO | SFT | GSPO | SFT | GSPO | SFT | GSPO |
| Higher permeability | 0.582 | **0.788** | 0.492 | **0.496** | 0.512 | **0.548** | 0.979 | **0.764** | 2.460 | **1.520** |
| Less like a drug | 0.816 | **0.944** | **0.598** | 0.438 | 0.792 | **0.892** | 0.990 | **0.443** | 2.488 | **1.422** |
| Less soluble in water | 0.802 | **0.940** | 0.574 | **0.902** | 0.722 | **0.934** | 0.898 | **0.262** | 1.880 | **0.289** |
| Less soluble in water + more HBA | 0.714 | **0.966** | **0.500** | 0.452 | 0.642 | **0.814** | 0.986 | **0.505** | 1.840 | **0.636** |
| Less soluble in water + more HBD | 0.780 | **0.880** | 0.700 | **0.794** | 0.728 | **0.826** | **0.995** | 1.000 | **2.841** | 2.893 |
| Lower permeability | 0.820 | **0.934** | 0.800 | **0.846** | 0.810 | **0.924** | **0.995** | 1.000 | **2.917** | 3.156 |
| More like a drug | 0.500 | **0.928** | **0.114** | 0.006 | **0.216** | 0.088 | 0.972 | **0.241** | 2.596 | **0.353** |
| More soluble in water | 0.788 | **0.920** | 0.588 | **0.842** | 0.752 | **0.912** | 1.000 | **0.996** | 3.530 | **3.491** |
| More soluble in water + higher permeability | **0.676** | 0.642 | **0.406** | 0.246 | **0.540** | 0.340 | 0.959 | **0.888** | 1.846 | **1.657** |
| More soluble in water + lower permeability | 0.800 | **0.946** | 0.602 | **0.826** | 0.752 | **0.936** | **0.998** | 1.000 | 2.558 | **2.575** |
| More soluble in water + more HBA | 0.772 | **0.930** | 0.560 | **0.864** | 0.724 | **0.918** | 0.987 | **0.981** | **2.591** | 2.701 |
| More soluble in water + more HBD | 0.836 | **0.964** | 0.584 | **0.890** | 0.786 | **0.952** | **0.995** | 1.000 | **2.711** | 2.720 |
| With more HBA | 0.806 | **0.976** | 0.792 | **0.900** | 0.792 | **0.900** | 0.995 | **0.553** | 2.921 | **0.805** |
| With more HBD | 0.812 | **0.968** | 0.802 | **0.958** | 0.802 | **0.958** | 1.000 | **1.000** | 3.367 | **3.700** |
| Overall | 0.750 | **0.909** | 0.579 | **0.676** | 0.684 | **0.782** | 0.983 | **0.755** | 2.629 | **2.001** |

pute Validity as the fraction of outputs that can be parsed and sanitized by RDKit; invalid generations are treated as failures for success metrics.

(ii) **Optimization Success.** For a property $p \in \mathcal{P}$, let $\Delta p = p(x_{\mathrm{out}}) - p(x_{\mathrm{src}})$ and let $s_{I,p} \in \{+1, -1\}$ denote the desired direction under instruction $I$ (increase or decrease). We report: (i) **Success (Loose)**: $s_{I,p}\Delta p > 0$ for all $p \in \mathcal{P}_I$; (ii) **Success (Strict)** (primary): $s_{I,p}\Delta p \geq \tau_p$ for all $p \in \mathcal{P}_I$, with thresholds $\tau_{\mathrm{LogP}}=0.5$, $\tau_{\mathrm{QED}}=0.1$, $\tau_{\mathrm{TPSA}}=10.0\,\text{Å}^2$, and $\tau_{\mathrm{HBA}}=\tau_{\mathrm{HBD}}=1$. For multi-objective conditions, all objectives must satisfy the criterion simultaneously.

(iii) **Property Stability (Off-target Drift).** Let $\mathcal{P}_{\neg I} = \mathcal{P} \setminus \mathcal{P}_I$ be the non-target properties. We flag a *shift violation* on $p \in \mathcal{P}_{\neg I}$ if $|\Delta p| \geq \tau_p$ (using the same $\tau_p$ as above). We report: (i) **Shift Rate**: the fraction of *valid* outputs with at least one shift violation; (ii) **Shift Avg**: the mean number of violated non-target properties per valid output.

### 4.4. Results

#### 4.4.1. RESULTS ON PROTEIN SEQUENCE OPTIMIZATION

**Efficacy of Trajectory-Based Editing.** On GFP, we evaluate two edit regimes: *replace-only* edits and the full atomic action space $\mathcal{A}$ (INSERT/DELETE/REPLACE). Following Section 4.3, we sample $N=100$ candidates per method from the same source sequence and report **Success**, **Uniqueness**, and **Novelty**; AAV uses the same full-action protocol.

**Improved diversity and novelty under replace-only edits.** In the replace-only setting (Table 5), multiple methods achieve moderate success, but differ substantially in diversity. While the Zero-Shot baseline attains $53/100$ success, it collapses to only 9 unique improved sequences (3 novel), suggesting highly repetitive generations. Vanilla

SFT improves diversity (39 unique improved) but still produces a large fraction of training-set mutations (Novelty 28/39). In contrast, *STRIDE* achieves slightly higher success ($61/100$) while producing substantially more unique and novel improvements (59 unique; 53 novel). Overall, explicitly modeling edit trajectories strengthens exploration without sacrificing optimization success.

**Benefits amplify for variable-length edits.** The full action space $\mathcal{A}$ (Table 3) is substantially more challenging due to variable-length transformations. Because the fluorescence oracle is trained primarily in a substitution regime, full-action results should be interpreted as oracle-based controllability stress tests rather than absolute fluorescence measurements. Here, Vanilla SFT degrades sharply ($42/100$ success; 30 unique; 14 novel), indicating difficulty in consistently producing beneficial edits beyond simple substitutions. By contrast, *STRIDE* remains robust and achieves the strongest overall results ($89/100$ success; 78 unique; 76 novel), highlighting that explicit trajectories are particularly valuable when the edit space becomes more combinatorial.

*Table 3.* Results on the Fluorescence Landscape Prediction dataset under the full INSERT/DELETE/REPLACE action space. Because the fluorescence oracle is trained primarily on substitution data, these numbers are best read as oracle-based controllability stress tests rather than absolute fluorescence measurements.

| Method | Success | Unique | Novelty |
|---|---|---|---|
| Random Perturbation | 5/100 | 5/5 | 3/5 |
| Zero-Shot | 54/100 | 8/54 | 4/8 |
| Edit Flow | 79/100 | 51/79 | 13/51 |
| Vanilla SFT | 42/100 | 30/42 | 14/30 |
| *STRIDE* | **89/100** | **78/89** | **76/78** |

*Table 4.* Attribution ablation and cross-protein evaluation.

*(a)* GFP attribution ablation. Replace-only edits.

| Method | Success | Novelty |
|---|---|---|
| Direct final-seq. | 55 | 23 |
| Struct. edits (no rationale) | 48 | 40 |
| Full *STRIDE* | **61** | **53** |

*(b)* AAV transfer evaluation. Full action space.

| Method | Success | Unique | Novelty |
|---|---|---|---|
| Vanilla SFT | 69 | 52 | 28 |
| Edit Flow | 52 | 33 | 20 |
| *STRIDE* | **73** | **60** | **35** |

*Table 5.* Results on the Fluorescence Landscape Prediction dataset under replace-only edit operations.

| Method | Success | Unique | Novelty |
|---|---|---|---|
| Random Perturbation | 13/100 | 13/13 | 12/13 |
| Zero-Shot | 53/100 | 9/53 | 3/9 |
| EvoDiff-38M FT | 47/100 | 38/47 | 24/38 |
| Edit Flow | 53/100 | 44/53 | 37/44 |
| Vanilla SFT | 55/100 | 39/55 | 28/39 |
| *STRIDE* | **61/100** | **59/61** | **53/59** |

**Executable edit trajectories vs. diffusion/flow baselines.** We further compare against discrete diffusion/flow baselines (EvoDiff and Edit Flow). In the *replace-only* regime (Table 5), *STRIDE* outperforms both EvoDiff (47/100) and Edit Flow (53/100) in Success. In the full action space (Table 3), Edit Flow attains competitive Success (79/100) but yields limited novelty (13/51), whereas *STRIDE* maintains both high Success and high Novelty (76/78). These results suggest that trajectory-conditioned LLM editing provides a favorable balance between optimization strength and exploration compared to diffusion/flow baselines.

**Attribution and cross-protein generalization.** We further isolate *where* the gains come from and *whether they transfer*. For attribution, we compare three variants on GFP (replace-only): direct final-sequence generation, structured edit tokens alone (no free-form rationale), and full *STRIDE*. Structured edit tokens alone already lift Novelty $23 \rightarrow 40$ over direct generation, and full *STRIDE* attains the best overall trade-off (Table 4a), indicating that the gain is not merely from emitting a longer free-form rationale but from the executable edit-trajectory interface. For transfer, the same recipe applied to AAV under the full action space again places *STRIDE* first on all three metrics (Table 4b), suggesting that the interface generalizes across protein landscapes rather than overfitting to GFP-specific oracle artifacts.

**Closed-loop iterative refinement.** *STRIDE* can be reused as an outer-loop refiner by feeding the best executed output of one round back as input. On GFP with a fixed 90-candidate budget, closed-loop refinement consistently lifts novelty over one-shot ($84.3\% \rightarrow 97.0\%$ replace-only; $94.7\% \rightarrow 97.2\%$ full action; Appendix E.4), providing inference-time depth/breadth control without retraining. Each round stays within the reliable short-horizon regime (95/100 exact execution at 1–3 edits; Appendix E.3); we report closed-loop refinement on GFP only.

**Specialist optimizers and long-horizon execution.** We also benchmark against task-specialist baselines under their own protocols. On an EVOLVEpro-style (Jiang et al., 2025) active-learning simulation on GFP/AAV with ESM2-650M embeddings (Appendix E.1), our LLM-based sampler is competitive with EVOLVEpro on Top-16 metrics on both GFP and AAV (mean and p90 hit-rate 1.00); EVOLVEpro is stronger on the single best queried variant. Under

a predictor-aligned GGS (Kirjner et al., 2024) comparison on GFP-TAPE (Appendix E.2), the LLM sampler attains higher oracle fitness across all gaps, while GGS explores a broader region. Exact trajectory execution drops sharply with horizon: 95/100 at 1–3 mutations vs. 26/100 at 3–10 on GFP (Appendix E.3), a limitation discussed in Section 5.

*Table 6.* Overall comparison of Qwen3-SFT and *STRIDE* variants on DrugAssist. Success reports strict / loose success rates, while shift reports violation rate / average count.

| Method | Success (S / L) ↑ | Shift (R / A) ↓ |
|---|---|---|
| Random Perturbation | 0.000 / 0.000 | 1.000 / 2.444 |
| Zero-Shot | 0.147 / 0.190 | 0.833 / 2.267 |
| Vanilla SFT | 0.629 / 0.745 | 0.948 / 2.379 |
| Vanilla GSPO | 0.653 / 0.785 | 0.865 / 2.382 |
| *STRIDE*-SFT | 0.579 / 0.684 | 0.983 / 2.629 |
| *STRIDE*-GRPO | 0.782 / 0.876 | 0.956 / 2.588 |
| *STRIDE*-GSPO | 0.676 / 0.782 | **0.755 / 2.001** |
| *STRIDE*-CISPO | **0.784 / 0.878** | 0.954 / 2.581 |

### 4.4.2. RESULTS ON MOLECULAR OPTIMIZATION

**Trajectory-conditioned editing improves controllability under instruction.** We evaluate instruction-conditioned molecular optimization on 500 source molecules across 14 objectives (7,000 instances total), reporting validity, strict/loose success, and non-target property shifts (Section 4.3). Table 2 reports per-task shift statistics, while Table 6 summarizes overall performance across methods.

**Direct generation is strong but incurs large off-target shifts.** Vanilla SFT (directly predicting the optimized SMILES without explicit trajectories) already achieves strong goal attainment (Strict/Loose: 0.629/0.745; Table 6), indicating that the Qwen3 backbone carries substantial chemical priors for plausible edits. However, it also exhibits considerable unintended property drift (Shift: 0.948/2.379), showing that high success does not necessarily imply controlled optimization.

**Explicit trajectories are a useful interface but require alignment.** Imposing an explicit, executable edit trajectory (*STRIDE*-SFT) introduces a stricter generation require-

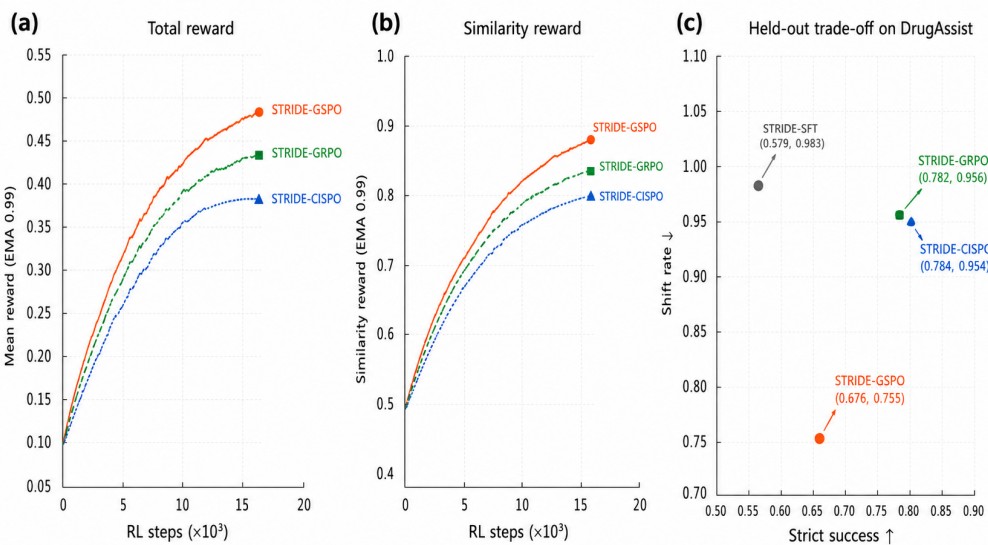

*Figure 2.* **(a,b)** Total and similarity reward vs. RL steps for *STRIDE*-GRPO/GSPO/CISPO (EMA, decay 0.99). **(c)** Held-out DrugAssist trade-off: strict success ($\uparrow$, $x$) vs. non-target shift rate ($\downarrow$, $y$); lower-right is best, *STRIDE*-SFT shown as reference.

ment and does not by itself improve stability (Strict/Loose: 0.579/0.684; Shift: 0.983/2.629). This suggests that trajectory supervision primarily provides a structured control interface—which is especially important under index-grounded, variable-length editing—but additional post-training is needed to align trajectories with the desired objective–stability trade-off.

**RL post-training.** Post-training *STRIDE* with policy optimization substantially improves objective satisfaction (Figure 2). *STRIDE*-GRPO and *STRIDE*-CISPO achieve the highest strict success (0.782 and 0.784), but their shift metrics remain high (Shift Rate $\approx$0.95). In contrast, *STRIDE*-GSPO attains the best overall controllability (Figure 2c; Shift: 0.755/2.001) while improving strict success over *STRIDE*-SFT (0.579 $\rightarrow$ 0.676) and substantially increasing validity (Table 2: 0.750 $\rightarrow$ 0.909). Compared to direct RL without trajectories (Vanilla GSPO), trajectory-conditioned RL yields a better stability profile (Shift Rate 0.865 $\rightarrow$ 0.755) at comparable success, showing that edit scripts are a more effective alignment substrate.

**Comparison with DrugAssist.** We compare against DrugAssist (Ye et al., 2025) on matched task families (Appendix E.5). After RL post-training, SMILES-GSPO surpasses DrugAssist on 4 of 6 tasks in strict success, with validity uniformly above 0.90.

## 5. Discussion

**Impact of Model Scale.** Model capacity matters for index-grounded, long-horizon editing: Qwen3-14B follows specified indices in $\sim$80% of actions, whereas Qwen3-4B-Thinking-2507 drops below 60% and often drifts after inser-

tions/deletions shift positions. This gap suggests that reliable executable editing requires enough capacity to maintain a mutable sequence state across multi-step trajectories.

**Protein GRPO is prone to mode collapse.** Protein GRPO training can drive the policy onto a small set of high-scoring trajectories (e.g., success $\sim$ 100/100 with unique improvements only $\sim$ 2/100); we therefore report Unique and Novelty alongside Success, since success alone can be vacuously satisfied by a degenerate policy.

**Limitations.** GFP full-action results are oracle-based controllability stress tests rather than absolute fluorescence, since the oracle is trained on substitutions; all gains are in-silico pending wet-lab validation, and GFP exact execution degrades with horizon length (95/100 at 1–3 edits vs. 26/100 at 3–10; Table 11). Future work includes indel-aware oracles, parser-in-the-loop correction, multi-anchor editing, and lighter adapted backbones.

## 6. Conclusion

We presented *STRIDE*, a post-training framework that casts discrete biological sequence optimization as trajectory-level planning over executable atomic edits (INSERT/DELETE/REPLACE). Pairing shortest-path supervision via Levenshtein backtracing with group-based reward alignment, *STRIDE* delivers consistent gains in novelty and controllability across GFP, AAV, and instruction-conditioned molecular editing, with closed-loop inference-time control. Explicit, parseable edit programs offer a productive interface for coupling LLMs with constrained, variable-length search problems in scientific design.

## Impact Statement

*STRIDE* produces explicit INSERT/DELETE/REPLACE trajectories that experts can audit before use. Candidates require wet-lab validation; we caution against using these methods to enhance pathogenicity or transmissibility. Used responsibly, it can accelerate transparent biological design.

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

# A. Edit Flows Baseline

We implement Edit Flows (Havasi et al., 2025) as a non-autoregressive discrete-flow baseline for variable-length protein sequence generation and editing. Unlike fixed-length masked discrete diffusion, Edit Flows defines a Continuous-Time Markov Chain (CTMC) directly over the space of sequences and parameterizes its generator via *edit operations* (INSERT/DELETE/REPLACE), naturally supporting length-changing transformations. We follow the training loss and first-order CTMC simulation procedure in (Havasi et al., 2025), with task-specific choices (protein vocabulary, edit budget, and length cap) described below.

## A.1. State Space and Edit Parameterization

Let $\mathcal{V}$ be the amino-acid vocabulary (plus special tokens such as BOS/EOS/PAD), and let $\mathcal{X} = \bigcup_{n=0}^{L_{\max}} \mathcal{V}^n$ be the space of sequences up to a maximum length $L_{\max}$. Given a current sequence $x_t \in \mathcal{X}$ and a normalized time $t \in [0, 1]$, the model outputs five heads:

- **Rate heads $\boldsymbol{\lambda}$**: three non-negative per-position intensities $\lambda_{\text{ins}}(x_t, t)_i$, $\lambda_{\text{del}}(x_t, t)_i$, $\lambda_{\text{sub}}(x_t, t)_i$ for insertion, deletion, and substitution.

- **Distribution heads $\mathbf{Q}$**: two categorical distributions $Q_{\text{ins}}(\cdot \mid x_t, t)_i$ and $Q_{\text{sub}}(\cdot \mid x_t, t)_i$ over $\mathcal{V}$ for sampling inserted/substituted tokens.

We apply a Softplus activation to ensure $\lambda \geq 0$.

**Masking constraints.** To preserve structural validity, we force $\lambda_{\text{ins}} = \lambda_{\text{del}} = \lambda_{\text{sub}} = 0$ on PAD positions. At the BOS position, we set $\lambda_{\text{del}} = \lambda_{\text{sub}} = 0$ to preserve the sequence anchor, while allowing insertion.

## A.2. Model Architecture

We use a Transformer Encoder to parameterize the time-dependent CTMC generator. The input is the current sequence $x_t$ (tokenized) and the scalar time $t$. We add a time embedding for $t$ to the token representations at each layer, together with learned positional embeddings.

## A.3. Training via Alignment-Conditioned Discrete Flow Matching

Training requires a supervision signal describing which edits would transform a partially noised sequence into the target. Given a paired example $(x_0, x_1)$, we compute a Levenshtein alignment and obtain aligned sequences $(z_0, z_1)$ in an augmented space that includes a special blank token $\epsilon$ (used only for defining the auxiliary alignment process, not part of $\mathcal{V}$).

**Stochastic interpolation.** We sample a time $t \sim \text{Uniform}(0, 1)$ and define a token-wise mixture schedule $\kappa(t) = t^{\kappa_{\text{pow}}}$ (we use $\kappa_{\text{pow}} = 3$). We sample an intermediate aligned sequence $z_t$ by taking each aligned position from $z_1$ with probability $\kappa(t)$ and from $z_0$ otherwise, and then strip blanks to obtain $x_t$.

**Loss.** The alignment $(z_t, z_1)$ specifies the set of *remaining* edit operations needed to transform $x_t$ into $x_1$: INSERT where $z_t = \epsilon$ and $z_1 \neq \epsilon$, DELETE where $z_t \neq \epsilon$ and $z_1 = \epsilon$, and SUBSTITUTE where both are non-blank but disagree. Following (Havasi et al., 2025), we use a Monte-Carlo estimate of the Edit Flows loss, which contains (i) a sum of all predicted rates and (ii) a weighted log-intensity term for the remaining edits:

$$\mathcal{L}(\theta; x_t, t) = \sum_i \left( \lambda_{\text{ins},i} + \lambda_{\text{del},i} + \lambda_{\text{sub},i} \right) - \frac{\dot{\kappa}(t)}{1 - \kappa(t)} \sum_{(i,\text{op}) \in \mathcal{E}(z_t, z_1)} \log r_\theta(\text{op}_i \mid x_t, t), \qquad (1)$$

where $\dot{\kappa}(t) = \frac{d\kappa(t)}{dt}$ and $r_\theta$ is the model-assigned intensity of the ground-truth edit: $r_\theta(\text{INS}_i = v) = \lambda_{\text{ins},i} Q_{\text{ins}}(v)_i$, $r_\theta(\text{DEL}_i) = \lambda_{\text{del},i}$, $r_\theta(\text{SUB}_i = v) = \lambda_{\text{sub},i} Q_{\text{sub}}(v)_i$.

## A.4. Sampling under an Edit Budget

We generate candidates by simulating the learned CTMC with the standard first-order approximation (Havasi et al., 2025). With step size $\Delta t$, for each position $i$ we sample:

- INSERT with probability $p_{\text{ins},i} = \text{clamp}(\lambda_{\text{ins},i}\Delta t, 0, 0.9)$;

- DELETE-or-SUBSTITUTE with probability $p_{\text{ds},i} = \text{clamp}((\lambda_{\text{del},i} + \lambda_{\text{sub},i})\Delta t, 0, 0.9)$. If triggered, we choose DELETE with probability $\lambda_{\text{del},i}/(\lambda_{\text{del},i} + \lambda_{\text{sub},i})$, otherwise SUBSTITUTE.

If INSERT or SUBSTITUTE happens at $i$, the new token is sampled from $Q_{\text{ins}}(\cdot)_i$ or $Q_{\text{sub}}(\cdot)_i$, respectively. All sampled edits are applied simultaneously.

To match our evaluation protocol, we enforce an atomic edit budget $B \in \{1, 2, 3\}$ by stopping the simulation once the number of executed atomic edits reaches $B$. To prevent runaway length growth, we additionally impose a hard cap $L_{\text{max}}$; if the sequence exceeds this length, it is truncated and subsequent insertions are disabled.

## B. R-Group vs. Index-Level SFT

Table 7 shows a trade-off between two editing interfaces for SFT on DrugAssist tasks: (i) **R-group-level** actions that modify molecules at the functional-group/substituent level, and (ii) **index-level (IDX)** actions that operate on token/atom-level positions. Overall, R-group editing achieves higher validity (0.827 vs. 0.750), while IDX achieves higher strict success (0.579 vs. 0.546) and slightly higher loose success (0.684 vs. 0.648). The advantage of IDX is most pronounced on tasks that benefit from fine-grained structural adjustments (e.g., *Lower permeability*: strict success 0.800 vs. 0.572), whereas R-group editing can be competitive or preferable on some objectives (e.g., *Higher permeability*: strict success 0.686 vs. 0.492). Unless otherwise stated, we use IDX as the default interface in the main experiments and report R-group results for completeness.

*Table 7.* Comparison of SFT (IDX) and SFT (R-group) on DrugAssist tasks. Success metrics are higher-is-better, while shift-related metrics are lower-is-better. Bold marks the better value per cell.

| Task | Valid ↑ | | Success (Strict) ↑ | | Success (Loose) ↑ | | Shift Rate ↓ | | Shift Avg ↓ | |
|---|---|---|---|---|---|---|---|---|---|---|
| | IDX | R | IDX | R | IDX | R | IDX | R | IDX | R |
| Higher permeability | 0.582 | **0.880** | 0.492 | **0.686** | 0.512 | **0.742** | 0.979 | **0.911** | 2.460 | **2.214** |
| Less like a drug | **0.816** | 0.706 | **0.598** | 0.450 | **0.792** | 0.596 | **0.990** | 0.992 | **2.488** | 3.127 |
| Less soluble in water | 0.802 | **0.838** | **0.574** | 0.542 | **0.722** | 0.660 | 0.898 | **0.945** | **1.880** | 2.196 |
| Less soluble in water + more HBA | 0.714 | **0.734** | **0.500** | 0.396 | **0.642** | 0.492 | 0.986 | **0.967** | **1.840** | 1.970 |
| Less soluble in water + more HBD | 0.780 | **0.870** | **0.700** | 0.512 | **0.728** | 0.626 | 0.995 | **0.986** | 2.841 | **2.756** |
| Lower permeability | **0.820** | 0.730 | **0.800** | 0.572 | **0.810** | 0.624 | **0.995** | 0.995 | 2.917 | **2.556** |
| More like a drug | 0.500 | **0.814** | 0.114 | **0.168** | 0.216 | **0.284** | **0.972** | 0.973 | **2.596** | 2.767 |
| More soluble in water | 0.788 | **0.870** | 0.588 | **0.614** | 0.752 | **0.764** | 1.000 | **0.998** | 3.530 | **3.343** |
| More soluble in water + higher permeability | 0.676 | **0.876** | **0.406** | 0.380 | 0.540 | **0.552** | 0.959 | **0.943** | **1.846** | 1.865 |
| More soluble in water + lower permeability | 0.800 | **0.856** | **0.602** | 0.550 | **0.752** | 0.724 | 0.998 | **0.993** | 2.558 | **2.437** |
| More soluble in water + more HBA | 0.772 | **0.830** | **0.560** | 0.548 | **0.724** | 0.686 | **0.987** | 1.000 | 2.591 | **2.511** |
| More soluble in water + more HBD | 0.836 | **0.908** | 0.584 | **0.716** | 0.786 | **0.810** | 0.995 | **0.989** | 2.711 | **2.641** |
| With more HBA | **0.806** | 0.758 | **0.792** | 0.656 | **0.792** | 0.656 | **0.995** | 0.997 | 2.921 | **2.765** |
| With more HBD | 0.812 | **0.910** | 0.802 | **0.856** | 0.802 | **0.856** | 1.000 | **0.991** | **3.367** | 3.396 |
| Overall | 0.750 | **0.827** | **0.579** | 0.546 | **0.684** | 0.648 | 0.983 | **0.977** | 2.629 | **2.613** |

## C. Training Details

We implement our models using the Qwen3-14B architecture. We use Qwen3's reasoning-region chat template to serialize the structured `edit_traj` block; the model emits the edit trajectory inside this region followed by the final sequence. Below we detail the hyper-parameters for both the Supervised Fine-Tuning (SFT) and Group Relative Policy Optimization (GRPO) stages.

**Optimizer & Regularization (SFT)**

- **Base Model:** Qwen3-14B

- **Optimizer:** AdamW

- **Precision:** bfloat16 (BF16)

- **Learning Rate:** $5 \times 10^{-5}$

- **Scheduler:** Cosine decay with 20 warmup steps

- **Per-Device Batch Size:** 4

- **Gradient Accumulation:** 2 steps

- **Epochs:** 5

- **Quantization:** 8-bit

- **Gradient Checkpointing:** Enabled

- **Random Seed:** 42

**LoRA Adapters (SFT)**

- **Rank ($r$):** 32

- **Alpha ($\alpha$):** 64

- **Dropout:** 0.05

- **Target Modules:** q_proj, k_proj, v_proj, o_proj, up_proj, down_proj

**Optimizer & Regularization (GRPO)**

- **Optimizer:** AdamW

- **Learning Rate:** $1 \times 10^{-5}$

- **Scheduler:** Cosine decay

- **Max Steps:** 25,000

- **Per-Device Batch Size:** 1

- **Gradient Accumulation:** 4

- **KL Control:** Fixed

- **KL Coefficient:** 0.001

**GRPO Algorithm Parameters**

- **Number of Generations:** 8

- **Clip Ratio ($\epsilon$):** 0.2

- **Reward Function:** Task-specific (Protein Fluorescence / Chemical Property)

**GSPO Algorithm Additional Parameters**

- **Epsilon:** $3 \times 10^{-4}$

- **Epsilon High:** $4 \times 10^{-4}$

- **Steps per Generation:** 4

- **Beta:** 0

**CISPO Algorithm Additional Parameters**

- **Epsilon High:** 5.0

**LoRA Adapters (GRPO)**

- **Rank ($r$):** 16
- **Alpha ($\alpha$):** 32
- **Target Modules:** q_proj, k_proj, v_proj, o_proj, up_proj, down_proj

**DeepSpeed & Hardware**

- **Strategy:** DeepSpeed ZeRO-2 (Stage 2)
- **Offload:** None
- **Contiguous Gradients:** Enabled
- **Gradient Clipping:** Auto

## D. Off-Target Property Drifts

Table 8 provides a per-property breakdown of off-target drifts across the 14 DrugAssist optimization tasks, complementing the aggregate shift metrics reported in Section 4.4.2. We report shift rates for five molecular properties—HBA, HBD, logP, QED, and TPSA—only when the corresponding property is not included in the task objective / reward; entries marked as "–" are excluded accordingly. Shift rates are computed over valid outputs only, using the same shift definition as in Section 4.4.2.

Overall, GSPO often reduces unintended changes in non-target properties compared to the SFT baseline, though not uniformly across all tasks and properties. For example, in the "Less like a drug" task, GSPO reduces off-target drift in HBA ($0.824 \rightarrow \textbf{0.383}$) and TPSA ($0.762 \rightarrow \textbf{0.358}$). Similarly, in a multi-objective setting such as "Less soluble in water + more HBA", GSPO maintains tighter control over unrelated properties such as TPSA ($0.908 \rightarrow \textbf{0.072}$). These results suggest that sequence-level policy optimization in GSPO improves robustness beyond the aggregate shift metrics by reducing several off-target drifts while optimizing for the target objective.

*Table 8.* Per-property off-target shift rates for *STRIDE*-SFT vs. *STRIDE*-GSPO on DrugAssist tasks; lower is better. "–" marks the task's target property (excluded from non-target shift). Bold marks the lower (better) value between SFT and GSPO per cell.

| Task | HBA Shift | | HBD Shift | | logP Shift | | QED Shift | | TPSA Shift | |
|------|-----|------|-----|------|-----|------|-----|------|-----|------|
|      | SFT | GSPO | SFT | GSPO | SFT | GSPO | SFT | GSPO | SFT | GSPO |
| Higher permeability | 0.667 | **0.373** | 0.460 | **0.391** | 0.794 | **0.487** | 0.540 | **0.269** | – | – |
| Less like a drug | 0.824 | **0.383** | **0.289** | 0.354 | 0.613 | **0.326** | – | – | 0.762 | **0.358** |
| Less soluble in water | 0.656 | **0.026** | 0.070 | **0.006** | – | – | 0.683 | **0.240** | 0.471 | **0.017** |
| Less soluble + more HBA | – | – | 0.207 | **0.135** | – | – | 0.725 | **0.429** | 0.908 | **0.072** |
| Less soluble + more HBD | **0.974** | 0.995 | – | – | – | – | **0.910** | 0.916 | **0.956** | 0.982 |
| Lower permeability | 0.993 | **0.970** | 0.563 | 0.887 | **0.656** | 0.732 | 0.705 | **0.567** | – | – |
| More like a drug | 0.748 | **0.075** | 0.532 | **0.034** | 0.652 | **0.196** | – | – | 0.664 | **0.047** |
| More soluble in water | **0.949** | 0.980 | **0.881** | 0.911 | – | – | 0.734 | **0.661** | 0.967 | **0.939** |
| More soluble + higher permeability | 0.553 | **0.477** | 0.740 | **0.726** | – | – | 0.553 | **0.455** | – | – |
| More soluble + lower permeability | **0.953** | 0.985 | **0.882** | 0.928 | – | – | 0.723 | **0.662** | – | – |
| More soluble + more HBA | – | – | **0.863** | 0.974 | – | – | 0.756 | **0.751** | **0.972** | 0.976 |
| More soluble + more HBD | **0.892** | 0.979 | – | – | – | – | 0.833 | **0.741** | **0.986** | 1.000 |
| With more HBA | – | – | 0.536 | **0.092** | 0.653 | **0.305** | 0.762 | **0.348** | 0.970 | **0.059** |
| With more HBD | **0.921** | 0.994 | – | – | **0.623** | 0.955 | 0.830 | **0.756** | **0.993** | 0.996 |

## E. Specialist Baselines, Long-Horizon Execution, and Backbones

This appendix contains the full numerical tables for the protein-side additions referenced in the main text: an EVOLVEpro-aligned active-learning comparison, a predictor-aligned GGS comparison, the long-horizon execution stress test, and a cross-backbone study of the *STRIDE* recipe.

### E.1. EVOLVEpro-Aligned Active-Learning Comparison on GFP and AAV

We instantiate the EVOLVEpro (Jiang et al., 2025) active-learning protocol on the same raw labeled candidate pools used for our GFP/AAV evaluation, treating each pool as a closed candidate universe. EVOLVEpro is configured with ESM2-650M embeddings, a random-forest surrogate, random initialization, and top-$N$ acquisition for 10 rounds with batch size 16 (160 total queries). Both methods are evaluated with the same EVOLVEpro metrics: best queried activity, Top-16 mean activity, and Top-16 p90 hit rate, all computed from ground-truth activity. EVOLVEpro numbers are averaged over 5 seeds; "Ours" is a single-run external LLM sampler over a fixed candidate pool re-ranked by the shared surrogate.

*Table 9.* Aligned comparison with EVOLVEpro on GFP and AAV under matched top-set metrics. Top-16 p90 is the fraction of Top-16 queries reaching the 90th-percentile activity.

| Dataset | Method | Best ↑ | Top-16 mean ↑ | Top-16 p90 ↑ |
|---|---|---|---|---|
| GFP | EVOLVEpro | **4.0057** $\pm 0.0607$ | 3.8395 $\pm 0.0401$ | 0.9812 $\pm 0.0593$ |
| GFP | Ours | 3.9531 | **3.8907** | **1.0000** |
| AAV | EVOLVEpro | **6.2025** $\pm 0.3365$ | 4.6258 $\pm 0.7150$ | 0.9875 $\pm 0.0395$ |
| AAV | Ours | 5.7031 | **4.7857** | **1.0000** |

### E.2. Predictor-Aligned GGS Comparison on GFP-TAPE

Following the GGS (Kirjner et al., 2024) protocol, both methods share the same gap-specific smoothed predictor for ranking, the same GFP-TAPE oracle for final evaluation, and the same definitions of mean fitness, diversity, and novelty. For each gap, we construct the GGS base pool from the bottom $30\%$ fitness slice while enforcing that sequences are at least the corresponding mutational distance away from the top $1\%$ high-fitness set. We then run the standard GGS pipeline (unsmoothed predictor $\rightarrow$ GS smoothing $\rightarrow$ smoothed predictor $\rightarrow$ GWG $\rightarrow$ oracle evaluation) with `tik-gamma-1`, `ham1_n-250K`, `GWG_MAX_EPOCHS=10`, and greedy top-128 evaluation, averaged over 4 GWG seeds. For our external LLM sampler, we start from $1,000$ raw generations, retain 995 valid length-237 sequences and 822 unique candidates after deduplication, then for each gap re-rank this same fixed candidate pool using the corresponding gap-specific smoothed predictor, evaluate the top-128 with the GFP-TAPE oracle, and compute novelty against the same gap-specific base pool.

*Table 10.* Predictor-aligned GGS comparison on GFP-TAPE across four gap settings. Our LLM sampler attains higher oracle fitness across all gaps, while GGS produces broader, more diverse exploration, particularly at lower gaps.

| Method | Gap | Fitness ↑ | Diversity | Novelty |
|---|---|---|---|---|
| GGS (smoothed, 4-seed mean) | gap1 | 0.1992 | 15.3707 | 5.3379 |
| GGS (smoothed, 4-seed mean) | gap2 | 0.2656 | 11.7402 | 4.4180 |
| GGS (smoothed, 4-seed mean) | gap3 | 0.4507 | 4.9241 | 3.1113 |
| GGS (smoothed, 4-seed mean) | gap6 | 0.6007 | 3.5145 | 4.9746 |
| External LLM sampler (fixed 822-candidate pool) | gap1 | **0.7766** | 3.8468 | 1.9766 |
| External LLM sampler (fixed 822-candidate pool) | gap2 | **0.7799** | 3.8463 | 1.9922 |
| External LLM sampler (fixed 822-candidate pool) | gap3 | **0.7576** | 3.8322 | 2.4219 |
| External LLM sampler (fixed 822-candidate pool) | gap6 | **0.7607** | 3.8346 | 4.8203 |

### E.3. Long-Horizon Execution Stress Test

We probe how reliably the model executes its own edit programs as the horizon grows by training SFT-only variants on GFP demonstrations binned by mutation count and measuring exact trajectory execution. Execution is highly reliable in the short-horizon regime but degrades sharply once the budget exceeds three mutations, consistent with the long-horizon limitation discussed in Section 5.

### E.4. Closed-Loop Iterative Refinement on GFP

We test whether the trained *STRIDE* checkpoint can be reused in an outer closed loop by feeding the best executed output of one round back as the next-round input. We fix the total sampling budget to 90 candidates and compare three schedules: one-shot $1\times90$, $2\times45$, and $3\times30$, in both the replace-only and full INSERT/DELETE/REPLACE edit regimes.

*Table 11.* Long-horizon execution on GFP. Exact trajectory execution rate as a function of mutation count for an SFT-only *STRIDE* variant.

| Mutation count | Exact execution success |
|---|---|
| 1–3 | **95/100** |
| 3–10 | 26/100 |

*Table 12.* Closed-loop iterative refinement on GFP under a fixed total budget of 90 candidates. Closed-loop schedules consistently lift Novelty and uniqueness over one-shot generation.

| Schedule | Improved | Unique improved | Novelty |
|---|---|---|---|
| *Replace-only* | | | |
| One-shot $1 \times 90$ | 51/90 | 50 | 84.3% |
| Closed-loop $2 \times 45$ | 66/90 | 63 | 95.5% |
| Closed-loop $3 \times 30$ | **67/90** | **66** | **97.0%** |
| *Full INSERT / DELETE / REPLACE* | | | |
| One-shot $1 \times 90$ | 75/90 | 65 | 94.7% |
| Closed-loop $2 \times 45$ | **77/90** | **73** | 96.1% |
| Closed-loop $3 \times 30$ | 72/90 | 71 | **97.2%** |

### E.5. Aligned Comparison with DrugAssist and SELFIES Variants

We compare *STRIDE* variants against the DrugAssist (Ye et al., 2025) baseline on six task families that have a clean mapping to a DrugAssist task. For each method we report validity ($\in [0, 1]$) and strict/loose success on a fixed held-out evaluation pool. SMILES-SFT and SMILES-GSPO denote our *STRIDE* pipeline with SMILES inputs; SELFIES-SFT replaces the SMILES grammar with SELFIES (Krenn et al., 2020) and applies SFT only.

*Table 13.* Comparison with DrugAssist on six task families. Each cell is "valid / strict / loose". Best strict success per row is in bold.

| DrugAssist task | *STRIDE* instruction | DrugAssist | SMILES-SFT | SMILES-GSPO | SELFIES-SFT |
|---|---|---|---|---|---|
| qed+ | More like a drug | 0.970 / 0.630 / 0.760 | 0.500 / 0.114 / 0.216 | 0.928 / 0.006 / 0.088 | **1.000 / 0.282 / 0.406** |
| bbbp+ | Higher permeability | 0.980 / 0.610 / 0.820 | 0.582 / 0.492 / 0.512 | 0.788 / 0.496 / 0.548 | **1.000 / 0.606 / 0.730** |
| solubility+ | More soluble in water | 0.980 / 0.410 / 0.800 | 0.788 / 0.588 / 0.752 | **0.920 / 0.842 / 0.912** | 0.988 / 0.688 / 0.792 |
| acceptor+ | With more HBA | 0.960 / 0.670 / 0.710 | 0.806 / 0.792 / 0.792 | **0.976 / 0.900 / 0.900** | 0.940 / 0.738 / 0.738 |
| donor+ | With more HBD | 0.950 / 0.760 / 0.720 | 0.812 / 0.802 / 0.802 | **0.968 / 0.958 / 0.958** | 0.962 / 0.838 / 0.838 |
| sol+&acc+ | More soluble + more HBA | 0.950 / 0.270 / 0.500 | 0.772 / 0.560 / 0.724 | **0.930 / 0.864 / 0.918** | 0.954 / 0.528 / 0.634 |

**SELFIES representation.** Replacing SMILES with SELFIES (Krenn et al., 2020) lifts validity to $\approx 1.0$ and yields the strongest *STRIDE* results on drug-likeness, permeability, and HBD, while SMILES-GSPO remains stronger on harder multi-objective settings where structured chemical editing offsets the harder grammar.

### E.6. Cross-Backbone Study

To examine how the *STRIDE* recipe interacts with model family and pre-training, we additionally train the two-stage *STRIDE* pipeline on Llama-3-8B and Phi-4-reasoning-plus-14B alongside our default Qwen3-14B backbone (Table 14). Qwen3-14B is the strongest of the three under *STRIDE*. Llama-8B sees a modest gain from *STRIDE* over vanilla SFT, while Phi-4-reasoning-plus-14B—which has a stronger vanilla baseline but does not support a customizable reasoning region—fails to benefit from the structured edit-trajectory interface and underperforms its vanilla counterpart. We conclude that the gains require both sufficient capacity and the ability to follow customizable structured reasoning instructions.

*Table 14.* Cross-backbone protein optimization results (GFP, replace-only). *STRIDE* consistently helps backbones that support a customizable reasoning region (Qwen3, Llama) but does not transfer to backbones with fixed reasoning behavior (Phi-4-reasoning-plus).

| Model | Success | Unique | Novelty |
|---|---|---|---|
| Qwen3-14B (vanilla) | 55 | 39 | 28 |
| Qwen3-14B (*STRIDE*) | **61** | **59** | **53** |
| Phi-4-reasoning-plus-14B (vanilla) | 61 | 55 | 40 |
| Phi-4-reasoning-plus-14B (*STRIDE*) | 32 | 27 | 8 |
| Llama-3-8B (vanilla) | 28 | 20 | 20 |
| Llama-3-8B (*STRIDE*) | 30 | 30 | 27 |

