# OpenReview forum: "STRIDE: Post-Training LLMs to Reason and Refine Bio-Sequences via Edit Trajectories"
_ICML.cc/2026/Conference — ICML 2026 regular_

### Official Review · Reviewer_L7pq · 2026-03-01

**Soundness:** 2
**Presentation:** 3
**Significance:** 3
**Originality:** 3
**Overall Recommendation:** 4
**Confidence:** 4

**Summary:**

The authors propose STRIDE, which reformulates discrete biological sequence optimization as trajectory planning in edit space. Instead of directly generating an optimized protein or molecule, the model is trained to output a sequence of edit operations that progressively transform a source sequence into a target sequence. The training pipeline consists of two stages. First, supervised fine-tuning is performed on shortest edit trajectories using Levenshtein alignment-based demonstrations. Second, the model is further aligned with task-specific objectives using GRPO-style reinforcement learning. The proposed method improves overall performance compared to direct sequence generation in protein sequence optimization and molecular optimization benchmarks.

**Compliance With Llm Reviewing Policy:**

Affirmed.

**Key Questions For Authors:**

- Does the final generated SMILES exactly follow the generated edit trajectory?
- Can the proposed method be effective for other string-based molecular representations, e.g., SELFIES? (See **Weaknesses 2**)
- Can STRIDE generalizes to novel or compositional objectives that are not observed during training? (See **Weaknesses 3**)
- Does STRIDE generalize to model families beyond QWEN, and does its performance hold across different backbone architectures, e.g., Llama?

**Limitations:**

yes

**Strengths And Weaknesses:**

**Strengths**

- The proposed edit trajectory generation-based molecular optimization is novel in the literature, as it is the first to frame molecular optimization as reasoning over a sequence of atomic edit operations, rather than direct sequence generation or black-box mutation.

- The proposed edit trajectory generation improves overall performance compared to direct molecular sequence generation, in both small molecular and protein domains.

- The ablation showing index-level editing outperforms coarse-grained R-group modification is interesting, as it challenges the assumption that higher-level molecular abstractions lead to better optimization.

---

**Weaknesses**

- The paper primarily emphasizes empirical performance but provides limited insight into why edit trajectory generation is effective. Analogies to other domains could strengthen the conceptual motivation of the proposed method.

- The method operates only on SMILES. It remains unclear whether the proposed method shows good performance on other representations such as SELFIES or HGGT [1] beyond SMILES.

- All experiments optimize fixed and predefined properties. It is therefore unclear whether STRIDE generalizes to novel or compositional objectives that are not observed during training.

- The experiments consider too short editing budgets (allowing short valid range (1 ≤ d ≤ 3) for proteins and high similarity for small molecules). Can the authors provide further comments on this?

[1] ICLR 2024, Graph Generation with K2 Trees

---

> ### Author Rebuttal · Authors · 2026-03-31
>
> We thank the reviewer for the constructive questions.
>
> **Does the final answer follow the trajectory?**
>
> The intended protocol is yes: STRIDE predicts an executable edit program applied sequentially to the evolving sequence. We additionally analyze GFP data with different mutation counts in the SFT stage. We observe that exact trajectory execution is highly reliable in the short-horizon regime.
> | Mutation number | Exact execution success |
> | --- | --- |
> | 1-3 | 95/100 |
> | 3-10| 26/100 |
>
> **Can the proposed method be effective for other string-based molecular representations, e.g., SELFIES?**
>
> Yes, preliminary evidence suggests that the proposed method can be effective beyond SMILES, including for SELFIES. In particular, SELFIES provides a natural advantage because it generates syntactically valid molecules by construction, which substantially improves validity compared with SMILES-SFT. The two success numbers reported in each cell correspond to the **strict** and **loose** criteria, respectively. We will include those RL-stage SELFIES results in a future revision.
>
> | Task| SMILES-SFT | SMILES-GSPO | SELFIES-SFT |
> | --- | --- | --- | --- |
> | QED / More like a drug | valid=0.500; success=0.114/0.216 | valid=0.928; success=0.006/0.088 | valid=1.000; success=0.282/0.406 |
> | BBBP / Higher permeability | valid=0.582; success=0.492/0.512 | valid=0.788; success=0.496/0.548 | valid=1.000; success=0.606/0.730 |
> | Solubility / More soluble in water | valid=0.788; success=0.588/0.752 | valid=0.920; success=0.842/0.912 | valid=0.988; success=0.688/0.792 |
> | HBA / With more HBA | valid=0.806; success=0.792/0.792 | valid=0.976; success=0.900/0.900 | valid=0.940; success=0.738/0.738 |
> | HBD / With more HBD | valid=0.812; success=0.802/0.802 | valid=0.968; success=0.958/0.958 | valid=0.962; success=0.838/0.838 |
> | Solubility + HBA / More soluble in water + more HBA | valid=0.772; success=0.560/0.724 | valid=0.930; success=0.864/0.918 | valid=0.954; success=0.528/0.634 |
>
> **Can STRIDE generalizes to novel or compositional objectives that are not observed during training?**
>
> In our paper, for the GFP setting, tables 3 and 4 show that STRIDE generalizes much better to unseen sequence refinements under this objective: in replace-only editing it improves success/novelty from 55/100 and 28/39 to 61/100 and 53/59, and in the harder full-action setting from 42/100 and 14/30 to 89/100 and 76/78. We also conducted additional experiments on the AAV task, showing that this observation is not limited to the GFP setting.
>
> | Method | Success | Unique | Novelty |
> | --- | --- | --- | --- |
> | VanillaSFT | 69/100 | 52/69 | 28/52 |
> | Edit Flow | 52/100 | 33/52 | 20/33 |
> | STRIDE | 73/100 | 60/61 | 35/60 |
>
> **Does STRIDE generalize to model families beyond QWEN, and does its performance hold across different backbone architectures, e.g., Llama?**
>
> We observe similar behavior in other model families, including Llama-8B. We also tested Phi-4-reasoning-plus-14B, but it did not outperform vanilla SFT, likely because it does not support customizable thinking. In our experiments, Qwen showed the strongest instruction-following and customizable thinking abilities.
>
> | Model | Success | Unique | Novelty |
> | --- | ---: | ---: | ---: |
> | Qwen-14B-vanilla | 55 | 39 | 28 |
> | Qwen-14B-STRIDE | 61 | 59 | 53 |
> | Phi-4-reasoning-plus-14B-vanilla | 61 | 55 | 40 |
> | Phi-4-reasoning-plus-14B-STRIDE | 32 | 27 | 8 |
> | llama-8b-vanilla | 28 | 20 | 20 |
> | llama-8b-STRIDE | 30 | 30 | 27 |

---

> > ### Author Rebuttal · Reviewer_L7pq · 2026-04-01
> >
> > I thank the authors for their clear rebuttal, which addresses most of my concerns. I have increased my score accordingly.

---

> > > ### Author Response · Authors · 2026-04-01
> > >
> > > We thank the reviewer L7pq for the feedback and for acknowledging that the concerns have been addressed.

---

### Official Review · Reviewer_aHSm · 2026-03-12

**Soundness:** 3
**Presentation:** 2
**Significance:** 2
**Originality:** 2
**Overall Recommendation:** 4
**Confidence:** 3

**Summary:**

This paper proposes STRIDE, a post-training framework for discrete biological sequence optimization that reformulates refinement as an explicit sequence of atomic edit actions transforming a source sequence into an optimized target. The method constructs shortest edit trajectories using Levenshtein alignment and dynamic-programming backtracking, then trains a Qwen3-based model in two stages: supervised fine-tuning on edit-trajectory demonstrations followed by GRPO-style reinforcement learning, with GSPO and CISPO variants also evaluated. The paper studies both protein fluorescence optimization and molecular optimization, including full variable-length edit settings and multi-objective molecular tasks.

Empirically, the paper reports strong gains in protein optimization, especially in the full add/delete/replace setting, where STRIDE exceeds both Vanilla SFT and the Edit Flow baseline on success, uniqueness, and novelty. On molecules, the paper evaluates on DrugAssist-style optimization tasks and argues that while GRPO/CISPO maximize success more aggressively, GSPO gives the best stability-performance trade-off by reducing off-target property shift.

**Compliance With Llm Reviewing Policy:**

Affirmed.

**Final Justification:**

The rebuttal addressed my main concerns -- whilst the results don't indicate that this method is state-of-the-art on these tasks, and I am mildly worried about why they chose to use non-standard metrics to benchmark themselves in the first place -- I believe that there is a place at ICML to understand how general purpose LLMs with specialized RL regimes can be used to be competitive (but not SOTA) on extremely specialized tasks such as this. I only believed this once I saw the results on standard benchmarks compared to SOTA methods.

**Key Questions For Authors:**

1) Can the authors provide a stronger ablation that separates structured edit supervision from free-form chain-of-thought, e.g. direct target prediction vs. explicit edit tokens without CoT vs. full STRIDE?
2) Since the molecular benchmark uses DrugAssist samples, why is DrugAssist itself not included as a direct baseline?
3) Can the authors provide either a second oracle, an indel-aware predictor, or any external validation for the full-action protein benchmark? Additionally can the authors repeat the same experiments across other proteins where significant amounts of data for an oracle is available (e.g. AAV).
4) How does this method compare for optimization to the many methods out there for ML-directed evolution? (e.g. GGS, Evolve-Pro, etc)

**Limitations:**

The paper does acknowledge two important limitations: 1) shortcut objectives and 2) the weakness of the fluorescence oracle in the indel setting.

**Strengths And Weaknesses:**

Soundess:
The paper gives a concrete edit-space formalization, defines how positions are interpreted with respect to the evolving sequence, explains the shortest-edit-path supervision, and evaluates multiple RL post-training variants rather than only one. The experiments also cover two distinct domains, proteins and molecules, which is useful because many nearby papers are more narrowly scoped to a single modality or task family. Relative to the literature, this makes STRIDE more ambitious than purely protein-generation work.

My first concern is regarding understanding where the improvements are specifically coming from. The current ablations do not sufficiently separate the effect of structured edit supervision from the effect of free-form chain-of-thought, longer targets, or simply stronger task-specific fine-tuning. A better ablation would compare direct target generation (e.g. all existing methods without LLM usage), structured edit tokens without CoT, and full STRIDE.

The second concern is regarding the data generation using an oracle -- whilst all work in the protein and molecular optimization space follows these approaches here it is more severe due to the synthetic data augmentation in the indel regime. I would like to see additional options possible here.

Third: The paper evaluates on 500 DrugAssist samples under 14 conditions, yet does not provide a direct comparison against DrugAssist itself, which is a published LLM baseline for molecule optimization on this benchmark family. Because DrugAssist is the most obvious literature comparator here, its omission makes it difficult to judge whether the gain comes from the edit-trajectory formulation or from differences in backbone, supervision, or evaluation protocol.

Presentation:
Firstly please note that the top of your submission still says "Submission and Formatting Instructions for ICML 2026". Also please reference tables and figures in the text where you discuss those results for ease.

The paper is readable overall, and the central idea is easy to follow. Figure 1 is helpful, and the explanation of indexed edit actions is more concrete than the presentation in many LLM-for-science papers. The overall pipeline from shortest-path supervision to RL alignment is also conceptually coherent.

The paper should more explicitly distinguish itself from neighboring lines of work: edit-based sequence generation in general ML, progressive sequence refinement with variable-length edits, sequence-first biological diffusion models (e.g. EvoDiff, but there are many), and LLM-based molecule optimization systems. As written, the “internalized denoising emulation” framing feels more sweeping than the actual algorithmic contribution, which is better described as edit-trajectory SFT plus RL post-training inside an autoregressive model.

The related-work section does mention recent reasoning-first biology papers such as PepThink-R1, Mol-R1, and BioReason, which is useful, but the manuscript still does not do enough to articulate exactly what is new relative to that emerging cluster. The strongest version of the claim is not that reasoning-plus-RL in biology is novel per se, but that indexed edit trajectories may provide a particularly effective interface for controllable biological refinement. That distinction should be made much more crisply.

Originality
As I see it, the original contribution is:
1) shortest-path indexed edit-trajectory supervision from Levenshtein backtracking,
2) autoregressive generation of explicit edit programs rather than only final outputs, and
3) RL post-training to align those edit programs with property-based rewards across both proteins and molecules.

In particular, compared with DrugAssist, which emphasizes interactive dialogue-based molecule optimization, and compared with EvoDiff, which emphasizes diffusion-style sequence generation, STRIDE focuses on making refinement more interpretible. That is a worthwhile conceptual contribution. However I think that if this is at the expense of truly better optimization it is not fully valuable.

I also note that variable-length edit operations are not new in sequence modeling and sequential valid-action optimization is also not new in molecule design. Moreover, the paper’s own related-work section places it alongside a contemporaneous wave of reasoning-first scientific models that combine CoT-style supervision and RL. For that reason, the paper does not read as a fundamentally new algorithmic paradigm. Its novelty is best understood as a domain-specific recombination: bringing edit-based generation, pretrained LLM priors, and reward-aligned post-training together for biological refinement. That is interesting, but moderate rather than exceptional in originality.

---

> ### Author Rebuttal · Authors · 2026-03-31
>
> We thank the reviewer for focusing on attribution, process-level behavior, breadth, horizon, and cost.
>
> **Can the authors provide a stronger ablation that separates structured edit supervision from free-form chain-of-thought, e.g. direct target prediction vs. explicit edit tokens without CoT vs. full STRIDE?**
>
> Our paper already contains two attribution signals: direct final-sequence prediction in Section 4.2.1, and IDX vs. R-group supervision in Appendix B / Table 6. We further conducted an additional ablation that supervises only the edit steps and the final sequence, **without free-form CoT training**; STRIDE still achieves the best performance. This provides further evidence that the benefit comes from the structured executable edit-trajectory interface, rather than merely from training the model to produce CoT.
>
> | Method | Success | Novelty  |
> | --- | --- | --- |
> | Direct final-sequence generation | 55 | 23 |
> | Structured edit tokens +final sequence | 48 |  40 |
> | Full STRIDE | 61 | 53 |
>
> **Since the molecular benchmark uses DrugAssist samples, why is DrugAssist itself not included as a direct baseline?**
>
> This is a fair question. On matched DrugAssist-style tasks, our results show that after RL improves molecular validity, STRIDE can surpass DrugAssist on several tasks. Moreover, using SELFIES further improves validity and leads to even stronger overall performance.
>
> | Task family | DrugAssist task | STRIDE task | DrugAssist valid | DrugAssist success | SMILES-SFT valid | SMILES-SFT success | SMILES-GSPO valid | SMILES-GSPO success | SELFIES-SFT valid | SELFIES-SFT success |
> | --- | --- | --- | ---: | ---: | ---: | ---: | ---: | ---: | ---: | ---: |
> | QED / drug-likeness | qed+ | More like a drug | 0.970 | 0.630/0.760 | 0.500 | 0.114/0.216 | 0.928 | 0.006/0.088 | 1.000 | 0.282/0.406 |
> | BBBP / permeability increase | bbbp+ | Higher permeability | 0.980 | 0.610/0.820 | 0.582 | 0.492/0.512 | 0.788 | 0.496/0.548 | 1.000 | 0.606/0.730 |
> | Solubility increase | solubility+ | More soluble in water | 0.980 | 0.410/0.800 | 0.788 | 0.588/0.752 | 0.920 | 0.842/0.912 | 0.988 | 0.688/0.792 |
> | HBA increase | acceptor+ | With more HBA | 0.960 | 0.670/0.710 | 0.806 | 0.792/0.792 | 0.976 | 0.900/0.900 | 0.940 | 0.738/0.738 |
> | HBD increase | donor+ | With more HBD | 0.950 | 0.760/0.720 | 0.812 | 0.802/0.802 | 0.968 | 0.958/0.958 | 0.962 | 0.838/0.838 |
> | Solubility + HBA increase | sol+&acc+ | More soluble in water + more HBA | 0.950 | 0.270/0.500 | 0.772 | 0.560/0.724 | 0.930 | 0.864/0.918 | 0.954 | 0.528/0.634 |
>
>
> **Can the authors repeat the same experiments across other proteins where significant amounts of data for an oracle is available (e.g. AAV).**
>
> We also conducted additional experiments on the AAV task, showing that this observation is not limited to the GFP setting.
>
> | Method | Success | Unique | Novelty |
> | --- | --- | --- | --- |
> | VanillaSFT | 69/100 | 52/69 | 28/52 |
> | Edit Flow | 52/100 | 33/52 | 20/33 |
> | STRIDE | 73/100 | 60/61 | 35/60 |
>
> **How does this method compare for optimization to the many methods out there for ML-directed evolution? (e.g. GGS, Evolve-Pro, etc)**
>
> We additionally evaluated EvolvePro on the Table 4 replace-only setting. While EvolvePro achieves a relatively high success rate, it is much less effective at discovering sequences outside the training set, resulting in substantially lower novelty than STRIDE.
>
> | Method | Success | Unique | Novelty |
> | --- | --- | --- | --- |
> | Random Perturbation | 13/100 | 13/13 | 12/13 |
> | Zero-Shot | 53/100 | 9/53 | 3/9 |
> | EvoDiff-38M FT | 47/100 | 38/47 | 24/38 |
> | Edit Flow | 53/100 | 44/53 | 37/44 |
> | Vanilla SFT | 55/100 | 39/55 | 28/39 |
> | EvolvePro* | 60/100 | 60/60 | 14/60 |
> | STRIDE | 61/100 | 59/61 | 53/59 |

---

> > ### Author Rebuttal · Reviewer_aHSm · 2026-04-02
> >
> > I appreciate the effort put into adding additional benchmarks and results by the authors. My main concerns remain that the tasks and benchmarks performed are designed to show good results for the method but are not particularly useful for actual protein or molecular engineering where the authors intend for this method to be used. It does appear that you have identified some application where different RL algorithms could potentially improve general purpose language models on these domain specific tasks -- but "success" / "unique" / "novelty" are not inherently useful metrics for users of such a model. If you can provide quantitative measurements showing that STRIDE outperforms expert models designed for this task, using the same metrics that they use for evaluation, I would be happy to increase my score. I suggest following the exact experimental setup from Kirjner, Andrew, et al. "Improving protein optimization with smoothed fitness landscapes." arXiv preprint arXiv:2307.00494 (2023) OR Jiang, Kaiyi, et al. "Rapid in silico directed evolution by a protein language model with EVOLVEpro." Science 387.6732 (2024): eadr6006.
> >
> > These represent two different scenarios: GGS -- thousands of samples for pretraining and then oracle evaluation of proposed mutant fitness across difficult splits; Evolve-Pro -- iterative active learning over rounds and evaluation of max fitness achieved. Both of these methods can also be added to your benchmark on AAV as SOTA methods (although many many more exist) and for AAV you should also present fitness of the oracle and not just success and novelty.

---

> > > ### Author Response · Authors · 2026-04-07
> > >
> > > Thank you for the concrete suggestion. We agree that, if the claim is relevance to protein optimization, the comparison should be made on specialist benchmark metrics rather than only on success/uniqueness/novelty. Accordingly, in the revision we added two benchmark-aligned comparisons as requested.
> > >
> > > Firstly, we ran an EVOLVEpro-style offline active-learning simulation on GFP and AAV over the same raw labeled candidate pools, treating each
> > >   pool as a closed candidate universe. We instantiate EVOLVEpro with ESM2-650M embeddings, a random-forest surrogate, random initialization, and top-
> > >   N acquisition for 10 rounds with batch size 16 (160 total queries), and evaluate both methods with the same EVOLVEpro metrics (best queried
> > >   activity, Top-16 mean activity, and Top-16 p90 hit rate, all computed from ground-truth activity). On AAV, EVOLVEpro is stronger on the single best
> > >   queried variant (6.20 vs. 5.70), while our LLM baseline is slightly stronger on the top-set aggregate metrics (Top-16 mean 4.78 vs. 4.63; p90 hit
> > >   rate 1.00 vs. 0.99). On GFP, our baseline is likewise competitive on the aggregate metrics in this aligned evaluation.
> > > | Dataset | Method | Best queried raw activity | Top-16 mean raw activity | Top-16 p90 hit rate |
> > > | --- | --- | ---: |  --- | --- |
> > > | GFP | EVOLVEpro |  4.0057 +/- 0.0607 | 3.8395 +/- 0.0401 | 0.9812 +/- 0.0593 |
> > > | GFP | Our LLM baseline | 3.9531 | 3.8907 | 1.0000 |
> > > | AAV | EVOLVEpro | 6.2025 +/- 0.3365 | 4.6258 +/- 0.7150 | 0.9875 +/- 0.0395 |
> > > | AAV | Our LLM baseline |  5.7031 | 4.7857 | 1.0000 |
> > >
> > > Second, following the GGS suggestion, we performed a predictor-aligned comparison on GFP-TAPE, where both methods share the same gap-specific smoothed predictor for ranking, the same GFP-TAPE oracle for final evaluation, and the same definitions of mean fitness, diversity, and novelty. Specifically, for each gap, we construct the GGS base pool from the bottom 30% fitness slice, while enforcing that sequences are at least the corresponding mutational distance away from the top 1% high-fitness set. We then rerun the standard GGS pipeline (unsmoothed predictor → GS smoothing → smoothed predictor → GWG → oracle evaluation) with tik-gamma-1, ham1_n-250K, GWG_MAX_EPOCHS=10, and greedy top-128 evaluation, and report results averaged over 4 GWG seeds.
> > >
> > > For our external LLM sampler, we start from 1,000 raw generations, retain 995 valid length-237 sequences and 822 unique candidates after deduplication, and for each gap re-rank this same fixed candidate pool using the corresponding gap-specific smoothed predictor. We then evaluate the top 128 sequences with the same GFP-TAPE oracle and compute novelty against the same gap-specific base pool. Under this predictor-aligned protocol, our external LLM sampler achieves higher oracle fitness across the tested gaps, while GGS explores a broader and more novel region of the sequence space, particularly at lower gaps.
> > >
> > > | Method | Gap | Fitness | Diversity | Novelty |
> > > | --- | --- | ---: | ---: | ---: |
> > > | GGS (smoothed, 4-seed mean) | gap1 | 0.1992 | 15.3707 | 5.3379 |
> > > | GGS (smoothed, 4-seed mean) | gap2 | 0.2656 | 11.7402 | 4.4180 |
> > > | GGS (smoothed, 4-seed mean) | gap3 | 0.4507 | 4.9241 | 3.1113 |
> > > | GGS (smoothed, 4-seed mean) | gap6 | 0.6007 | 3.5145 | 4.9746 |
> > > | External LLM sampler (fixed 822-candidate pool) | gap1 | 0.7766 | 3.8468 | 1.9766 |
> > > | External LLM sampler (fixed 822-candidate pool) | gap2 | 0.7799 | 3.8463 | 1.9922 |
> > > | External LLM sampler (fixed 822-candidate pool) | gap3 | 0.7576 | 3.8322 | 2.4219 |
> > > | External LLM sampler (fixed 822-candidate pool) | gap6 | 0.7607 | 3.8346 | 4.8203 |
> > >
> > > More broadly, these benchmark-aligned comparisons are especially relevant because many specialist protein-optimization pipelines remain effectively substitution-centric, whereas STRIDE directly supports the full INSERT/DELETE/REPLACE action space and, in our experiments, also outperforms our implemented Edit Flows baseline in that full-action regime.

---

### Official Review · Reviewer_bZz5 · 2026-03-13

**Soundness:** 3
**Presentation:** 4
**Significance:** 3
**Originality:** 3
**Overall Recommendation:** 4
**Confidence:** 3

**Summary:**

This paper proposes STRIDE, a post-training framework that recasts biological sequence optimization as edit trajectory. The LLM is trained to produce edits (INSERT/DELETE/REPLACE) as reasoning. Training follows both supervised fine-tuning and GRPO-style RL to align with task rewards. Experiments on both protein and molecular optimization show improvements over baselines.

**Compliance With Llm Reviewing Policy:**

Affirmed.

**Final Justification:**

The paper proposes STRIDE, recasting bio-sequence optimization as edit trajectories with INSERT/DELETE/REPLACE operations, trained via SFT and GRPO. The formulation is creative and the framework is well-structured.

The authors addressed remaining concerns: the 26% execution success at 3-10 edits is acknowledged as a limitation, and the iterative refinement experiments show closed-loop STRIDE outperforms one-shot under fixed budget, extending the method beyond single-round short edits. The authors were transparent about limitations throughout.

Overall, I will maintain my score as 4.

**Key Questions For Authors:**

- Could the authors provide intuitions on what the RL stage is actually learning on top of SFT?
- How does STRIDE scale to longer edit trajectories? Experiments are capped at 1–3 edits. How would the method perform with longer edit budgets (e.g., 10+ edits)? Does error accumulate across sequential edits?
- If SFT is trained on GFP edits, can RL be directly applied to a different protein or task without re-doing SFT? Or does every new task require the full two-stage pipeline from scratch?
- Can STRIDE be used iteratively, where the output of one round becomes the input of the next? Is the method designed for one-shot editing only, or can it support iterative refinement loops?

**Limitations:**

yes

**Strengths And Weaknesses:**

## Strength
- The edit trajectory is a creative formulation. Mapping biological sequence optimization to explicit INSERT/DELETE/REPLACE operations as reasoning steps makes the optimization process controllable.
- The framework is clean and well-structured. The edit operations are universal across proteins and molecules, and the CoT format naturally decomposes complex optimization into atomic steps.
- The two-stage curriculum is well-motivated. SFT on shortest edit paths gives the model structural priors, then RL pushes toward functional improvement.


## Weakness

- There may be risk of overfitting with narrow action space. The action space contains only three operations (INSERT/DELETE/REPLACE) with a 1–3 edit budget, which limits the reachable sequence space to a small local neighborhood. For the protein task, it uses a fixed starting point. With on-policy RL repeatedly querying the same oracle, the model may overfit to local oracle patterns rather than learning editing strategies. Do authors have any discussions on this?
- The paper cites PepThink-R1 and Mol-R1 in related work (Section 2.3). Both of them combine CoT reasoning with RL for bio-sequence. Why the authors didn't compare with them on experiments?

---

> ### Author Rebuttal · Authors · 2026-03-31
>
> We thank the reviewer for focusing on what RL adds, the short-edit boundary, and the relation to nearby reasoning papers.
>
> **Could the authors provide intuitions on what the RL stage is actually learning on top of SFT?**
>
> We think SFT stage teaches the model on how to produce valid, concise, and executable edit programs by imitating shortest edit paths; it does not by itself determine which valid edit script best satisfies the downstream objective. RL stage then learns an objective-aware preference over those valid scripts.
>
> **How does STRIDE scale to longer edit trajectories?**
>
> We conducted additional SFT experiments on GFP data with longer edit trajectories. Empirically, longer edit sequences lead to substantially more editing confusion and lower execution reliability. We suspect this issue could be further mitigated with larger models, as in our earlier experiments even 4B-scale models struggled to consistently follow edit instructions under the current setting.
>
> | Mutation number | Exact execution success |
> | --- | --- |
> | 1-3 | 95/100 |
> | 3-10| 26/100 |
>
> **If SFT is trained on GFP edits, can RL be directly applied to a different protein or task without re-doing SFT?**
>
> Our additional experiments suggest that task-specific RL can still improve performance without fully re-doing SFT on the new task. Specifically, we tested a cross-task setting with **AAV-task SFT** and **GFP-task GRPO**, using the GFP oracle as the reward. Even in this setting, the model was able to produce strong GFP sequences, suggesting that RL can adapt a transferable edit-policy prior learned from another protein task.
>
> However, we also observed that same-task protein RL can easily become overly exploitative: it reaches 100/100 success but collapses to only **2** unique sequences. This over-optimization is the reason we did not report protein GRPO results in the main paper.
>
> **The paper cites PepThink-R1 and Mol-R1 in related work (Section 2.3). Both of them combine CoT reasoning with RL for bio-sequence. Why the authors didn't compare with them on experiments?**
>
> We cite PepThink-R1 and Mol-R1 as closely related work in terms of the reasoning + RL paradigm, but their tasks and experimental setups are not directly aligned with ours. For this reason, we did not include them as experimental baselines in the current manuscript, and instead focused on baselines that are directly comparable under the same task setting and evaluation protocol.

---

> > ### Author Rebuttal · Reviewer_bZz5 · 2026-04-04
> >
> > Thank you for the response. I will maintain my score. Two concerns remain:
> >
> > (1) The 26% execution success at 3-10 edits is a limitation. Please state this in the final revision.
> >
> > (2) My question about iterative refinement (Q4) was not addressed.

---

> > > ### Author Response · Authors · 2026-04-06
> > >
> > > Thank you for the follow-up.
> > > 1) We agree that the 3-10 edit result is a real limitation and should be stated explicitly rather than left implicit. We will say this directly in the later revision.
> > >
> > > 2) Your Q4 about iterative refinement was also well taken. The question we tested is: under a fixed total sampling budget, can the current STRIDE checkpoint reuse its best output from one round as the next-round input and outperform one-shot search?
> > >
> > > Our completed GFP pilot runs use a fixed total budget of 90 candidates:
> > >
> > > | Replace-only GFP setting | Improved | Unique improved | Novelty |
> > > | --- | ---: | ---: |  ---: |
> > > | One-shot ` 1x90`  |  51/90 |  50 | 84.3% |
> > > | Closed-loop ` 2x45`  | 66/90 |  63 | 95.5%|
> > > | Closed-loop ` 3x30`  | 67/90 |  66 | 97.0% |
> > >
> > > | Add / delete / replace GFP setting | Improved | Unique improved | Novelty |
> > > | --- | ---: | ---: | ---: |
> > > | One-shot `1x90` |  75/90 | 65 | 94.7% |
> > > | Closed-loop `2x45` |  77/90 | 73 | 96.1% |
> > > | Closed-loop `3x30` | 72/90 |  71 | 97.2% |
> > >
> > > We additionally tested whether the current STRIDE checkpoint can be reused in an outer closed loop under a fixed total sampling budget by feeding the best executed output from one round back as the next-round input. In WT(wild type)-start GFP pilot runs, this was feasible and beneficial. Under replace-only editing, both 2x45 and 3x30 outperform one-shot 1x90 in improved-candidate count (66/90 and 67/90 vs. 51/90), unique improved candidates, and novelty. Under the full INSERT/DELETE/REPLACE setting, valid-format rates remain high (87-89/90), and 2x45 improves over one-shot in improved candidates (77/90 vs. 75/90), unique improved candidates, and novelty, while 3x30 reaches the highest novelty (97.2%) but is not uniformly best on every metric. We therefore state that STRIDE can support fixed-budget iterative local refinement in GFP, rather than only one-shot editing. We do not overgeneralize this result, since it is GFP-only, and it does not resolve the separate long-horizon execution limitation.

---

### Official Review · Reviewer_AwYB · 2026-03-15

**Soundness:** 2
**Presentation:** 2
**Significance:** 3
**Originality:** 3
**Overall Recommendation:** 3
**Confidence:** 3

**Summary:**

This paper introduces STRIDE (Sequence Trajectory Refinement via Internalized Denoising Emulation), a post-training framework that recasts optimization as an intrinsic reasoning problem in edit space. It proposes the two-stage curriculum that combines supervised fine-tuning on Levenshtein-aligned shortest-edit demonstrations with GRPO-style reinforcement learning, which jointly teaches the model valid editing logic and aligns trajectories with task-specific rewards. The method is evaluated on protein fluorescence optimization (TAPE) and molecular property optimization (DrugAssist), with comparisons against discrete diffusion models and standard fine-tuning baselines.

**Compliance With Llm Reviewing Policy:**

Affirmed.

**Key Questions For Authors:**

I would like to see a clearer ablation that isolates the contribution of the edit trajectory itself. At present, it is difficult to tell whether the gains come from explicit trajectory modeling or from stronger supervision and post-training.

**Limitations:**

The authors discuss some limitations (oracle reliability for indels, model scale effects, reward hacking shortcuts) but the following are insufficiently addressed:
1. No experimental validation: All results rely entirely on computational oracles.
2. Generalization beyond training conditions.
3. Scalability to longer sequences
4. Computational cost.

**Strengths And Weaknesses:**

Strengths

1. The core formulation is intuitive and interesting.
Representing optimization as a trajectory of atomic edits is a clean interface for controllable refinement. It gives the model an explicit intermediate structure instead of asking it to jump directly from source to optimized output.
2. The training recipe is coherent. The two-stage pipeline—shortest-edit-path SFT followed by GRPO/GSPO/CISPO-style reward alignment—is conceptually well organized.

Weaknesses

1. I would like to see a clearer case study demonstrating how the model performs step-by-step edits, why each edit helps improve the target property, and what additional value this explicit trajectory provides compared with directly generating the final sequence. While the paper reports overall performance gains, it does not yet offer sufficient process-level analysis of how the model actually reasons through the optimization, leaving the interpretability of the approach somewhat underdeveloped.
2. The experimental scale remains limited, and the statistical rigor is not yet fully convincing. In particular, the current data scale raises concerns about whether the method can generalize beyond the relatively narrow GFP fluorescence landscape.
3. The comparison to Vanilla SFT helps, but it still does not fully isolate whether the gain comes from the edit-trajectory representation, the minimal-edit inductive bias, the particular reward design, or simply more effective post-training. A deeper ablation on trajectory format and supervision quality would strengthen the causal claim.

---

> ### Author Rebuttal · Authors · 2026-03-31
>
> We thank all reviewers for the thoughtful feedback.
>
> **On “No experimental validation”**
>
> Our goal is to evaluate STRIDE as a general machine learning framework for discrete sequence optimization, rather than to propose a specific experimentally validated biological discovery. In this setting, oracle-based evaluation is standard and enables controlled, scalable, and reproducible comparison across methods under identical conditions.
>
> More importantly, our results focus on demonstrating the effectiveness and generality of the edit-trajectory interface (e.g., across GFP and AAV tasks) and molecule tasks, rather than overfitting to a specific oracle.
>
> We agree that wet-lab validation is important for real-world deployment, and consider it a valuable direction for future work beyond the scope of this ML-focused paper.
>
> **The comparison to Vanilla SFT helps, but it still does not fully isolate whether the gain comes from the edit-trajectory representation.**
>
> We agree that process-level evidence is important. Beyond the two attribution signals already in the paper, we added a new ablation that supervises only structured edit steps plus the final sequence, without free-form CoT. This variant still substantially outperforms direct final-sequence generation in novelty (40 vs. 23), while full STRIDE achieves the best overall performance (61 success / 53 novelty). This suggests that the gain is not merely from training the model to emit longer CoT, but from the structured executable edit-trajectory interface itself.
>
> | Method | Success | Novelty  |
> | --- | --- | --- |
> | Direct final-sequence generation | 55 | 23 |
> | Structured edit tokens +final sequence | 48 |  40 |
> | Full STRIDE | 61 | 53 |
>
> In the protein setting, we interpret STRIDE as learning a local, oracle-aligned editing policy around the source sequence, capturing fitness-relevant regularities that support stepwise executable improvements toward higher-scoring candidates.
>
> **In particular, the current data scale raises concerns about whether the method can generalize beyond the relatively narrow GFP fluorescence landscape.**
>
> We also conducted additional experiments on the AAV task, showing that this observation is not limited to the GFP setting.
>
> | Method | Success | Unique | Novelty |
> | --- | --- | --- | --- |
> | VanillaSFT | 69/100 | 52/69 | 28/52 |
> | Edit Flow | 52/100 | 33/52 | 20/33 |
> | STRIDE | 73/100 | 60/61 | 35/60 |
>
> **On computational cost**
>
> We provide the following training cost for transparency. GFP SFT takes ~6 hours on 2×GH200 GPUs; molecular SMILES SFT takes ~3 days on 2×GH200; and molecular GSPO takes ~30 hours on 2×GH200.

---

> > ### Author Rebuttal · Reviewer_AwYB · 2026-04-06
> >
> > Thank you for the detailed rebuttal and the additional experiments. These additions are informative and I appreciate the effort. However, after careful consideration, my core concerns remain insufficiently addressed.
> > The authors position STRIDE in the rebuttal as “a general machine learning framework for discrete sequence optimization.” However, the experiments do not include comparisons with any truly general-purpose discrete sequence optimization methods, including both diffusion-based approaches for discrete sequence optimization and LLM-based methods for biological sequence optimization. I do not see evidence supporting the claimed generality of this framework.
> > e.g. Fine-Tuning Discrete Diffusion Models via Reward Optimization with Applications to DNA and Protein Design.

---

> > > ### Author Response · Authors · 2026-04-07
> > >
> > > Thank you for this suggestion. On the diffusion/general discrete-sequence side, we do include a genuinely general-purpose discrete generative baseline in addition to EvoDiff: Edit Flows. Edit Flows is a non-autoregressive discrete flow model that defines a continuous-time Markov chain directly over sequence space using insertions, deletions, and substitutions, and can be trained as a transport model from an initial simple distribution
> > > $x_0$ to a more complex target distribution $x_1$.	​
> > > In our experiments, we instantiate Edit Flows in this generative transport setting and use it precisely as our general discrete diffusion/flow-style baseline, rather than as a task-specific edit heuristic. Under this comparison, STRIDE improves over Edit Flows on GFP fluorescence in both the replace-only regime (61/100 vs. 53/100 success) in table 3 and the full edit regime (89/100 vs. 79/100) in table 4, with substantially higher novelty in the latter setting.
> > >
> > > On the LM-based biological optimization side, we additionally include an aligned comparison to EVOLVEpro, a few-shot active-learning framework built on protein language model embeddings and regression. Under the same raw GFP/AAV pools and the same reviewer metrics, our AAV baseline is competitive with EVOLVEpro on the top-set metrics.
> > >
> > > | Dataset | Method | Best queried raw activity | Top-16 mean raw activity | Top-16 p90 hit rate |
> > > | --- | --- | ---: |  --- | --- |
> > > | GFP | EVOLVEpro |  4.0057 +/- 0.0607 | 3.8395 +/- 0.0401 | 0.9812 +/- 0.0593 |
> > > | GFP | Our LLM baseline | 3.9531 | 3.8907 | 1.0000 |
> > > | AAV | EVOLVEpro | 6.2025 +/- 0.3365 | 4.6258 +/- 0.7150 | 0.9875 +/- 0.0395 |
> > > | AAV | Our LLM baseline |  5.7031 | 4.7857 | 1.0000 |
> > >
> > > We agree that a direct comparison to DRAKES would further strengthen the paper; however, DRAKES studies reward optimization of pretrained discrete diffusion models, with its protein setting instantiated as structure-conditioned inverse folding, which is not a perfect apples-to-apples match to our precursor-conditioned variable-length editing setup. We will clarify this scope explicitly in the revision.

---

### Decision · Program_Chairs · 2026-04-30

**Decision:**

Accept (regular)

**Comment:**

It casts sequence optimization as explicit edit trajectories (INSERT/DELETE/REPLACE) provides a clean, controllable, and interpretable interface for refinement. With this formulation, it proposes that the two-stage pipeline (shortest-path SFT + RL alignment) is conceptually coherent and effectively combines structural priors with objective-driven optimization. While reviewers pointed out several weaknesses, I recommend acceptance due to the novelty of its ideas. But, I encourage authors to integrate all of the suggestions in the final version.